# From Prompts to Responses: Dual-Sided Data Leakage and Defense in Split Large Language Models

Zixuan Gu[1]  Xiaojun Ye[1]  Yang Liu[2]

## Abstract

Large language models (LLMs) are increasingly deployed in privacy-sensitive domains, where users must balance the risk of data exposure through external APIs against the high computational cost of local deployment. Split learning has therefore emerged as a promising paradigm for LLM fine-tuning and inference under limited local resources. However, it introduces new privacy risks. Prior work primarily studies leakage of private input prompts, typically via inversion attacks on intermediate representations, while the potential for sensitive information leakage through generative response outputs remains largely unexplored.

In this work, we unveil novel vulnerabilities of Split-LLM by presenting **P**atched Model **I**nversion with **D**ual-Sided **I**nitialization(**PIDI**), a two-stage attack that simultaneously targets both private input prompts and output responses in Split-LLM settings. It combines dual-sided initialization with a patched inversion strategy to tackle long sequences, substantially outperforming prior inversion methods. To counter threats from both sides, we further propose the **A**dapter-based **D**ualGuard with **M**utual **I**nformation Defense(**ADMI**), which integrates an adapter-based local warmup strategy and mutual information regularization to provide a strong empirical privacy protection with minimal impact on task performance. Extensive experiments across diverse tasks and models demonstrate that ADMI effectively defends against PIDI and other state-of-the-art inversion attacks. Our code is publicly available at https://github.com/FLAIR-THU/VFLAIR-LLM.

[1]School of Software, Tsinghua University, Beijing, China [2]the Hong Kong Polytechnic University, Hongkong, China. Correspondence to: Yang Liu <yang-veronica.liu@polyu.edu.hk>.

*Proceedings of the $43^{rd}$ International Conference on Machine Learning*, Seoul, South Korea. PMLR 306, 2026. Copyright 2026 by the author(s).

## 1. Introduction

With the advancement of Large Language Models (LLMs), their applications have expanded into a growing number of fields. Privacy-sensitive domain users, however, face limitations in using external LLM APIs, while fully private deployments incur substantial computational costs. Split learning has then emerged as a privacy-preserving LLM fine-tuning and inference paradigm under resource constraints, termed as Split-LLM (Gu et al., 2025; Thapa et al., 2022; Lin et al., 2024; Shen et al., 2023).

In a typical Split-LLM, a full LLM model is divided into three slices, with the head and tail residing with private data holders and the body hosted on a cloud server. Despite this, such split settings remain vulnerable to input data leakage from the model head (Gu et al., 2025; Chen et al., 2024; Lin et al., 2024; Shen et al., 2023), as well as label leakage for classification tasks at the tail(Zhu et al., 2019; Li et al., 2022; Zou et al., 2022). While some recent works begin to examine the output response leakage (Liu et al., 2025; Fu et al., 2022) in open-ended generation, a systematic understanding of *how information can leak jointly from both model ends during generation* remains largely unexplored. This dual-ended leakage is particularly critical in high-stakes domains such as finance, where both input prompts and output responses contain highly sensitive information, such as a company's investment strategy or proprietary insights.

In this work, we study data leakage of both input prompts and generated responses in Split-LLM generation. Our main contributions are:

- We identify new vulnerabilities in Split-LLM and propose **P**atched Model **I**nversion with **D**ual Branch **I**nitialization(**PIDI**), a novel dual-sided attack that reconstructs both input prompts and end responses during open-ended generation by exploiting information leakage from both model ends.

- We propose **A**dapter-based **D**ualGuard with **M**utual Information Defense(**ADMI**) to mitigate threats from both ends, which simultaneously guard attacks from both ends and achieves a superior privacy-utility trade-off compared to other baselines.

## 2. Related Works

### 2.1. Split Learning of Large Language Models(Split-LLM)

To enable the private deployment of large language models (LLMs) under constrained local resources and data privacy, Split Learning(SL)(Gupta & Raskar, 2018; Thapa et al., 2022; Gu et al., 2025) has emerged as a promising solution. In the Split-LLM paradigm, a model is partitioned into multiple segments that are distributed between the client and the server, reducing the client's local computational burden and enhancing privacy by transmitting only intermediate activations or gradients rather than raw data. The "Head-Body-Tail (HBT)" Split-LLM(Lin et al., 2024; Gu et al., 2025) splits the LLM backbone into 3 slices, aiming to protect both private inputs and private outputs by isolating both ends of the model from the server and preventing direct exposure of sensitive data.

### 2.2. Data Attacks and Defenses in Split-LLM

Although transmitting only intermediate activations helps obscure private data, prior work shows that reconstruction attacks can still infer sensitive inputs from smashed representations. For example, (Pan et al., 2020; Song & Raghunathan, 2020; Qu et al., 2025) perform direct smashed data matching, while (Balunović et al., 2022; Deng et al., 2021) leverage backward gradient matching during training. (Chen et al., 2024) further combines forward and backward matching in the training phase with a novel SIP initialization. Beyond input leakage, (Zhu et al., 2019; Zou et al., 2022; Li et al., 2022) show that classification labels can be inferred from transmitted gradients. More recently, (Fu et al., 2022; Liu et al., 2025) further expand to generation tasks and demonstrate that private responses can be approximated by applying a pre-trained model tail to the model body's output activations.

To mitigate these threats, prior work injects noise via differential privacy or sparsification (Du et al., 2023; Chatzikokolakis et al., 2013a; Mai et al., 2023; Aji & Heafield, 2017), applies token-wise MLDP-based text perturbation (Yue et al., 2021; Chen et al., 2023; Tong et al., 2024), or introduces regularization-based objectives such as mutual information minimization(Zou et al., 2023; Duan et al., 2023), adversarial training(Pan et al., 2020), and prototype clustering (Zhou et al., 2023) to encourage privacy-preserving representations. These defenses primarily operate on the model head to prevent input prompt leakage, while providing limited protection against attacks that exploit the model tail. More recently, DualGuard (Liu et al., 2025) further adopts a local warm-up paradigm to induce rapid parameter shifts in both the model head and tail, aiming to jointly mitigate leakage from both sides. However, its warm-up stage replaces the model body with a lightweight projection layer

and subsequently freezes the model head during full training, which may constrain the extent of model adaptation and limit its protection ability.

Recently, VFLAIR-LLM (Gu et al., 2025) provides an open-source benchmark and research framework for systematically evaluating privacy risks and defenses in Split-LLM settings. Our work is implemented on top of this library.

## 3. Problem Definition

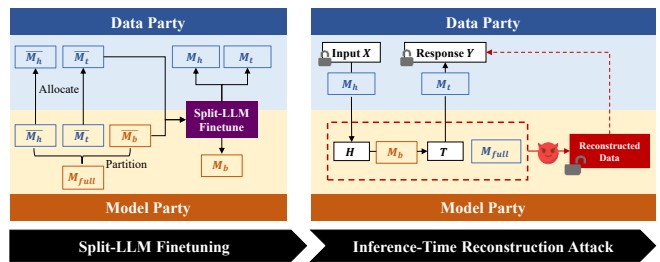

*Figure 1.* Dual Data Reconstruction Attack Threat Model

**The Split-LLM Setting.** We consider the typical head-body-tail(HBT) Split-LLM scenario(Gu et al., 2025; Chen et al., 2024), where two parties, a **Model Party** and a **Data Party**, collaborate to fine-tune an LLM model. The Model Party (e.g., an LLM provider with substantial computational resources) controls a full pre-trained LLM, denoted as $M_{\text{full}}$, which is partitioned into three distinct segments: (1) $\bar{M}_{\text{h}}$: The embedding layer $f_e$ and the initial $n_h$ decoders/encoders layers ; (2) $\bar{M}_{\text{b}}$: The intermediate $n_b$ layers, constituting the majority of the model's parameters; (3) $\bar{M}_{\text{t}}$: The final $n_t$ decoder/encoder layers and the output projection lm head $f_{lm}$. The Data Party (i.e., a private data holder with limited local resources) is allocated the lightweight $\bar{M}_{\text{h}}$ and $\bar{M}_{\text{t}}$ slices. The largest segment, $\bar{M}_{\text{b}}$, remains on the Model Party's server.

For an input sequence of $n$ tokens $X = [a_1, a_2, \ldots, a_n]$ and its generated response of $m$ tokens $Y = [a_{n+1}, a_{n+2}, \ldots, a_{n+m}]$, the auto-regressive generation requires $m$ collaborative Split-LLM forward passes at inference time. The system first processes the full input to produce the first output token $a_{n+1}$. In subsequent passes, the latest generated token is passed as input to the Split-LLM system due to the Key-Value (KV) Cache mechanism. By aggregating the transmitted activations across all $m$ passes, the Model party can obtain the model head's output activations $H = [h_1, h_2, \ldots, h_L] \in \mathbb{R}^{L \times d}$ and the model body's output activations $T = [t_1, t_2, \ldots, t_L] \in \mathbb{R}^{L \times d}$, where $d$ is the model's hidden dimension and $L = n + m - 1$.

Normally, the data party first employs **Split-LLM fine-tuning** to adapt a pre-trained model $\bar{M}_{\text{h/b/t}}$ using its private data. The fine-tuned model $M_{\text{h/b/t}}$ is then deployed via the **Split-LLM inference** pipeline to generate private responses

for data party's private input prompts.

**Threat Model** In this work, we assume the model party is an honest-but-curious attacker, which complies with the Split-LLM protocols and does not collude with external parties. Given a private input query $X = [a_1, a_2, \ldots, a_n]$ and the Split-LLM generated response $Y = [a_{n+1}, a_{n+2}, \ldots, a_{n+m}]$, the adversary is assumed to have access to the following information for conducting attacks during **Split-LLM inference**:

- The original model segments $\bar{M}_{h/b/t}$ along with the fine-tuned body $M_b$.

- The intermediate activations $H$ output by the model head and the corresponding outputs of the model body $T$.

- (Optional) A small amount of auxiliary data drawn from a distribution similar to that of the Data Party's private dataset, which can be used for training relevant attack models like in (Chen et al., 2024; Fu et al., 2022).

Leveraging this information, the Model Party seeks to reconstruct both the Data Party's **private inputs $X$ and generated responses $Y$ at inference time during autoregressive generation**. This setting differs from training-time attacks like(Chen et al., 2024; Balunović et al., 2022), as only forward computation is performed and *no gradients are available* for exploitation.

## 4. PIDI: Attacking Split-LLM Through Both Ends

By exploiting the auto-regressive nature of text generation, where generated tokens are iteratively appended to the input sequence, we design a **two-stage attack** to reconstruct private inputs and responses using both the transmitted hidden states $H$ and $T$, named **P**atched **M**odel **I**nversion with **D**ual-Sided **I**nitialization(**PIDI**). As described in Figure 2, in the first "Dual-Sided Initialization(DSI)", we construct an initial estimate of the private input and responses by jointly applying model completion and SIP technique on the input branch and the response branch, respectively. In the subsequent "Patched Model Inversion(PMI)" stage, we further refine this prior estimation via a novel patched model inversion training method to produce optimized embedding vectors, from which the final reconstructed sequence is extracted.

### 4.1. Dual-Sided Initialization(DSI)

In this stage, we first initialize our estimations of both input and response sequences through separate attacks, due

to their distinct characteristics. Specifically, we perform response and input initializations as follows:

- **Response Initialization via Model Completion (MC)**: Since the response sequence is closely tied to the model's generative patterns, we initialize it using **Model Completion (MC)**(Fu et al., 2022; Liu et al., 2025). This method leverages the 'not-too-far' property, where the parameters of the fine-tuned $M_t$ remain relatively close to those of the original pre-trained $\bar{M}_t$. The attacker can simply feed the model body's output $T$ into the original $\bar{M}_t$ to estimate the private response as $\hat{Y} = \bar{M}_t(T) = [a_{n+1}^0, a_{n+2}^0, \ldots, a_L^0]$, where $a_i^0$ denotes the initialized token at position $i$.

- **Input Initialization via SIP**: In contrast, the input sequence is weakly correlated with the model's generative process and is primarily encoded through the computational behavior of $M_h$. Hence, we initialize it using **Semi-white-box Forward Inversion Paradigm(SIP)**(Chen et al., 2024), which exploits the semantic properties of the hidden states $H$ and the known transformation $H = M_h(X)$. Using a small auxiliary dataset (50 samples in our experiments), we train a SIP model $M_{\text{SIP}}$ to directly estimate the private input from $H$ as $\hat{X} = M_{\text{SIP}}(H) = [a_1^0, a_2^0, \ldots, a_n^0]$.

To this end, the complementary strengths of MC and SIP enable us to construct a robust joint initialization $E_0 = [a_1^0, a_2^0, \ldots, a_L^0]$. However, these estimations remain coarse, as reflected by their limited attack performance in Figure 2. On the response side, discrepancies between $M_t$ and $bar M_t$, together with token sampling during generation, introduce systematic deviations from the true private responses. On the input side, prediction error of the SIP model further incurs wrong reconstruction. Consequently, the initialized sequence still needs further refinement. In the next stage, we further introduce a subsequent **Patched Model Inversion (PMI)** method to jointly optimize and refine these estimations.

### 4.2. Patched Model Inversion(PMI)

In this phase, we propose a patched model inversion strategy, extending the traditional model inversion method to better handle long text input sequences in transformer-based model scenarios.

Initially, the embedding estimations $E_0 \in \mathbb{R}^{L \times d}$ are fed into the original model head $\bar{M}_h$ to produce dummy hidden states $\hat{H} \in \mathbb{R}^{L \times d}$. Leveraging the *'not-too-far' property* between $\bar{M}_h$ and its fine-tuned counterpart $M_h$, the attacker refines $E_0$ by minimizing the inversion loss:

$$\mathcal{L}_{\text{inv}} = \|M_h(E) - H\| \quad E_1 = \arg\min_{E_0} \mathcal{L}_{\text{inv}} \quad (1)$$

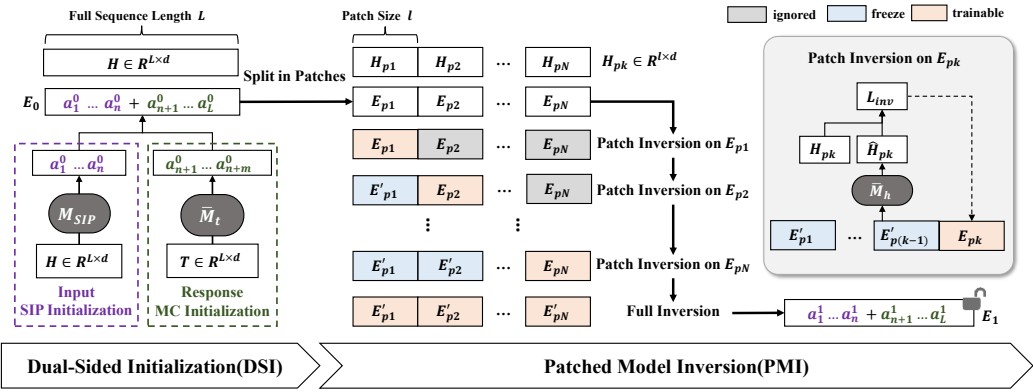

*Figure 2.* The Two-Stage Attack Methodology of PIDI

The resulting $E_1$ is then decoded to recover the private token sequence. However, directly optimizing embeddings of length $L$ becomes increasingly difficult as $L$ grows.

To mitigate this issue, we split the full embedding $E_0 \in \mathbb{R}^{L \times d}$ into $N$ non-overlapping patches of length $l$, denoted as $[E_{p1}, E_{p2}, \ldots, E_{pN}]$ where each patch $E_{pk} \in \mathbb{R}^{l \times d}$. The corresponding model head output $H$ can also be partitioned as: $H = [H_{p1}, H_{p2} \ldots H_{pN}]$, $H_{pk} \in \mathbb{R}^{l \times d}$, $\hat{H} = [\hat{H}_{p1}, \hat{H}_{p2}, \ldots \hat{H}_{pN}]$, $\hat{H}_{pk} \in \mathbb{R}^{l \times d}$. Due to the causal structure of decoder-only large language models, the prefix embeddings $[E_{p1}, E_{p2}, \ldots, E_{pk}]$ and the corresponding $[\hat{H}_{p1}, \hat{H}_{p2}, \ldots \hat{H}_{pk}]$ and $[H_{p1}, H_{p2}, \ldots, H_{pk}]$ can be treated as valid input–output pairs for the inversion problem, as the hidden states at these positions are independent of subsequent tokens. As described in Figure 2, we first conduct inversion training iteratively over the patches. At iteration $k$, only $E_{pk}$ is optimized by minimizing $\|\hat{H}_{pk} - H_{pk}\|$ while previous patches remain frozen. After individual refinement, all patches are unfrozen for a final joint inversion stage to improve overall refinement.

Results in Table 1 demonstrate that PIDI reconstructs private inputs and responses with high fidelity, posing a significant threat to both model ends. It highlights the urgent need for effective defenses. We now present our defense design, which mitigates such reconstruction attacks while preserving model utility.

# 5. ADMI: Defending LLMs From Private Data Leakage At Both Ends

To protect both the model head and tail, DualGuard (Liu et al., 2025) first establishes a "Local Warm-up + Full Training" paradigm. In the Local Warm-up, $M_h$ and $M_t$ are jointly trained via a projection layer with defense regularizers, aiming to break the "not-too-far" property and hinder model completion attacks. Full Training then continues with $M_h$ frozen. However, this approach has two limitations: (1) replacing $M_b$ with a simple projection layer can degrade

representations and final performance; (2) since the method does not change the system structure, the large $M_b$ can pull parameters back during full training, leaving $M_t$ vulnerable and the "not-too-far" property largely preserved.

To address these issues, we further propose **ADMI** (**A**dapter-based **D**ualGuard with **M**utual **I**nformation Defense):

- In Section 5.1, we introduce the two defense regularizers used in ADMI: **mutual information regularizer** $\mathcal{L}_{MI}$, which protects $M_h$ by hindering input leakage from $H$; and **model distance regularizer** $\mathcal{L}_D$, which breaks the "not-too-far" property to defend $M_t$ against model completion attacks. These regularizers define the core defensive objectives that guide parameter updates during training. These regularizers define the core defensive objectives guiding parameter updates.

- In Section 5.2, we describe the two-stage training of ADMI. First, the **Adapter-based Local Warm-up** introduces a novel adapter module into the system, enabling rapid parameter shifts guided by $\mathcal{L}_D$ while preserving performance. Second, the **Full Training** phase further adapts the model to downstream tasks, where both $\mathcal{L}_{MI}$ and $\mathcal{L}_D$ are applied to enforce dual end defense throughout the pipeline.

## 5.1. Dual Defense Regularizers

Apart from the main task loss $\mathcal{L}_T$, we design two defense regularizers, $\mathcal{L}_{MI}$ and $\mathcal{L}_D$, to protect the model head and tail respectively.

### 5.1.1. MUTUAL INFORMATION REGULARIZER $\mathcal{L}_{MI}$

To prevent $H$ from leaking information about the input $X$, we aim to minimize the mutual information $I(X, H)$, thereby limiting task-irrelevant input information retained in $H$. This increase the difficulty of reconstructing $X$ from $H$, hindering both the inversion training in PMI and the SIP initialization in DSI.

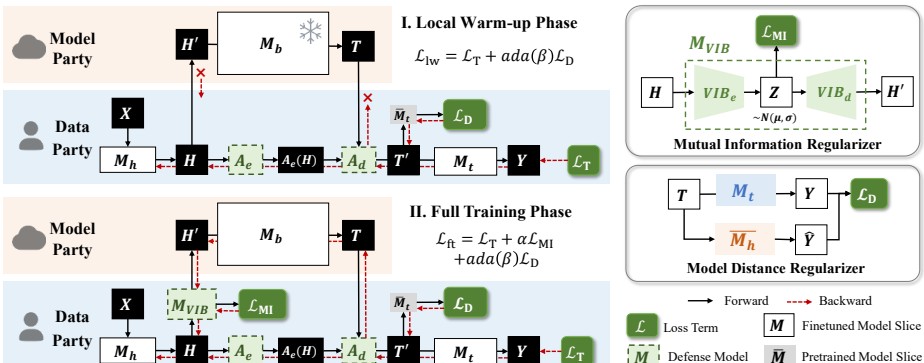

*Figure 3.* Defense Pipeline of ADMI.

Directly minimizing $I(X, H)$ is intractable due to the discrete and non-differentiable nature of the token ids $X$. We therefore operate on the continuous embedding representation $E = f_e(X)$. Since the embedding layer $f_e$ is typically fixed during fine-tuning, it forms a deterministic function. The Markov chain $X \to E \to H$ holds, and we come to $I(X, H) \leq I(E, H)$ by the data processing inequality. The objective can thus be reformulated as minimizing $I(E, H)$.

As the exact computation of $I(E, H)$ is infeasible, we adopt a variational upper bound based on the Variational Information Bottleneck (VIB) framework(Alemi et al., 2016; Zou et al., 2023). As illustrated in Figure 2, we insert a encoder-decoder stochastic bottleneck model $M_{\text{VIB}}$ between $M_{\text{h}}$ and $M_{\text{b}}$, whose encoder projects $H$ to Gaussian parameters ($\boldsymbol{\mu}$, $\boldsymbol{\sigma}$) and samples a latent variable via the reparameterization trick as $Z = \boldsymbol{\mu} + \boldsymbol{\sigma} \odot \boldsymbol{\epsilon}, \boldsymbol{\epsilon} \sim \mathcal{N}(0, I)$. The decoder then maps $Z$ to the perturbed representation $H'$, which is finally fed to $M_{\text{b}}$.

Following the derivation provided in Appendix A, the mutual information regularization term $\mathcal{L}_{\text{MI}}$ is defined as a tractable upper bound on $I(E, H')$ as in Equation (2):

$$\mathcal{L}_{\text{MI}} = \text{UpperBound}(I(E, H'))$$
$$= \frac{1}{N} \sum_{i=1}^{N} \frac{1}{2} \sum_{j=1}^{d} \left( \mu_{ij}^2 + \sigma_{ij}^2 - \log(\sigma_{ij}^2) - 1 \right) \quad (2)$$

where $N$ is the batch size and $d$ is the model's hidden dimension. The final training objective incorporates this term with a weighting coefficient $\alpha$, yielding the final mutual information regularization term as $\alpha \cdot \mathcal{L}_{\text{MI}}$.

### 5.1.2. MODEL DISTANCE REGULARIZER $\mathcal{L}_{\text{D}}$

To defend model completion attacks targeting $M_{\text{t}}$, we adopt the distance regularizer $\mathcal{L}_{\text{D}}$ proposed in (Liu et al., 2025), as illustrated in Figure 2. The objective is to maximize the output divergence between the fine-tuned model $M_{\text{t}}'$ and its pre-trained counterpart $M_{\text{t}}$, thereby breaking the "not-too-far" property to hinder model completion. The model distance loss $\mathcal{L}_{\text{D}}$ is defined as:

$$\mathcal{L}_{\text{D}} = \frac{1}{\text{CrossEntropy}(M_{\text{t}}(T), M_{\text{t}}'(T))} \quad (3)$$

To integrate this regularizer into the overall training objective, we weight it by a coefficient $\beta$.

### 5.2. Two-Stage Defense Training of ADMI

#### 5.2.1. ADAPTER-BASED LOCAL WARMUP

Excluding the central $M_{\text{b}}$ from the local warm-up phase may significantly disrupt model training. To address this limitation, we integrate an encoder–decoder Adapter module into the SL-LLM framework, as illustrated in Figure 3. The workflow and training dynamics are detailed below:

- During **forward propagation**, the adapter encoder $A_e$ first processes the original model head output $H$ into $A_e(H)$. Subsequently, $M_{\text{b}}$ receives the model head output $H$ and generates $T$. This representation $T$ is then passed to the Adapter decoder $A_d$, along with $A_e$ as memory to produce a refined representation $T'$. Finally, $T'$ is fed into $M_{\text{t}}$ to get the final result $Y$.

- During **backward propagation**, the Data Party does not transmit any gradients to the Model Party. As a result, $M_{\text{b}}$ *remains frozen* and only perform forward calculation, while the gradients of the main task loss $\mathcal{L}_{\text{T}}$ flow through $A_d$ and $A_e$ back to $M_{\text{h}}$, enabling its updates.

The Model Distance Regularizer $\mathcal{L}_{\text{D}}$ is added to the main task loss $\mathcal{L}_{\text{T}}$, with an additional $\beta$ controlling its weight. Therefore, the overall training objective during Local Warm-up can be expressed as:

$$\mathcal{L}_{\text{lw}} = \mathcal{L}_{\text{T}} + ada(\beta) * \mathcal{L}_{\text{D}} \quad (4)$$

In this way, the adapter module provides an alternative pathway for gradient flow, allowing us to leverage $M_{\text{b}}$'s comprehension ability while keeping it frozen, thereby preserving the local warm-up scheme's security properties.

*Table 1.* Attack Performance($AP_{\alpha=0.5}$) of the evaluated attack methods. Full results of $AP_{input}$ and $AP_{response}$ are placed in Tables 5 and 6.

| | Fin | | | Med | | | Dolly | | |
|---|---|---|---|---|---|---|---|---|---|
| | Llama3.2-3B | Llama3-8B | Qwen2.5-7B | Llama3.2-3B | Llama3-8B | Qwen2.5-7B | Llama3.2-3B | Llama3-8B | Qwen2.5-7B |
| PIDI(DSI+PMI) | **0.868** | **0.883** | **0.892** | 0.901 | **0.881** | **0.801** | **0.876** | **0.934** | **0.985** |
| DSI+VMI | 0.828 | 0.716 | 0.77 | **0.925** | 0.707 | 0.741 | 0.801 | 0.775 | 0.814 |
| MC+VMI | 0.43 | 0.423 | 0.442 | 0.373 | 0.264 | 0.424 | 0.473 | 0.454 | 0.412 |
| BiSR(SIP+VMI) | 0.665 | 0.610 | 0.546 | 0.851 | 0.546 | 0.706 | 0.483 | 0.323 | 0.667 |
| DSI | 0.691 | 0.697 | 0.77 | 0.686 | 0.707 | 0.706 | 0.707 | 0.691 | 0.74 |
| SIP | 0.391 | 0.347 | 0.546 | 0.58 | 0.537 | 0.667 | 0.341 | 0.307 | 0.45 |
| MC | 0.4 | 0.418 | 0.442 | 0.236 | 0.261 | 0.245 | 0.323 | 0.391 | 0.405 |
| VMI | 0.006 | 0.312 | 0.224 | 0.005 | 0.277 | 0.054 | 0.004 | 0.002 | 0.029 |

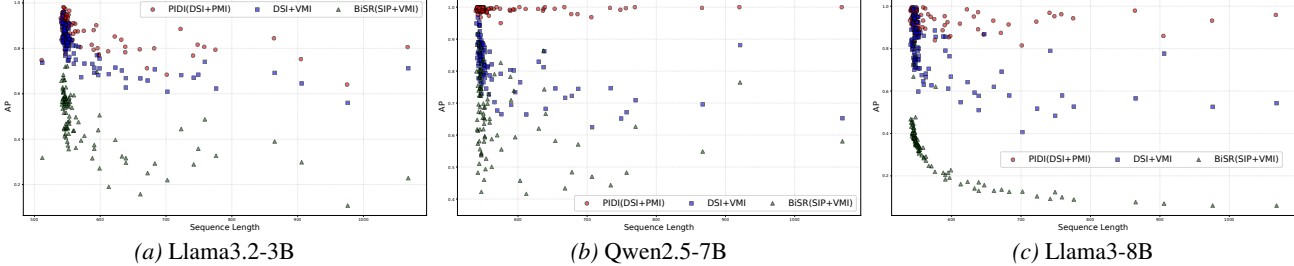

*(a)* Llama3.2-3B     *(b)* Qwen2.5-7B     *(c)* Llama3-8B

*Figure 4.* AP Across Varying Sequence Length (Evaluated on Dolly Dataset)

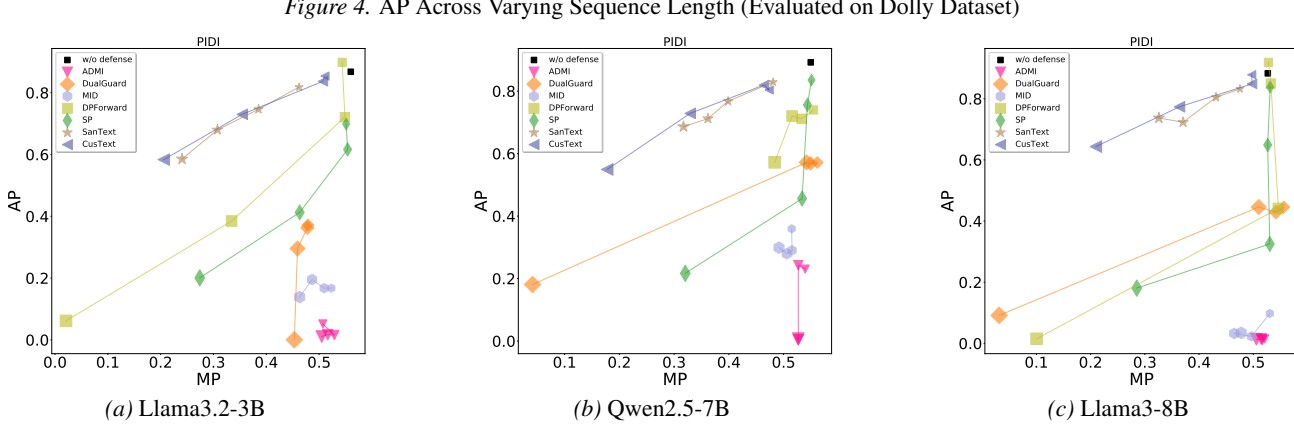

*(a)* Llama3.2-3B     *(b)* Qwen2.5-7B     *(c)* Llama3-8B

*Figure 5.* Defense Performance(MP-AP) for defending PIDI attack on the Fin Dataset. Dot size represents the defense strength. Here AP denotes $AP_{\alpha=0.5}$.

Considering the definition in Equation (3),$\mathcal{L}_D$ can be excessively large at early stages when $M_t(T)$ is close to $M'_t(T)$. Therefore, we adopt an **adaptive weight strategy** $ada(\beta)$ to balance it with the main task loss $\mathcal{L}_T$. Specifically, we compute the $\ell_2$ norms of their gradients w.r.t. input embeddings and define:

$$ada(\beta) = \frac{\min\left(\|\nabla_{\mathbf{E}}\mathcal{L}_D\|_2,\ 0.1 \cdot \|\nabla_{\mathbf{E}}\mathcal{L}_T\|_2\right)}{\|\nabla_{\mathbf{E}}\mathcal{L}_D\|_2 + \epsilon} \quad (5)$$

where $\epsilon$ is a small constant for numerical stability. This approach bounds the influence of $\mathcal{L}_D$, ensuring stable training and preserving utility.

### 5.2.2. FULL TRAINING WITH DEFENSE REGULARIZERS

After performing the aforementioned Local Warm-up Training, we then come to the second phase: Full Training. In this phase, Data Party and Model Party perform gradient communication, enabling gradients of the main task loss $\mathcal{L}_T$

to flow through the whole system. An additional Variational Information Bottleneck $M_{VIB}$ is inserted between $M_h$ and $M_b$ as detailed in Section 5.1.1. $\mathcal{L}_{MI}$ generated from $M_{VIB}$ together with $\mathcal{L}_D$ is added to main task loss $\mathcal{L}_T$, forming the overall objective as:

$$\mathcal{L}_{ft} = \mathcal{L}_T + \alpha\mathcal{L}_{MI} + ada(\beta) * \mathcal{L}_D \quad (6)$$

Here, $\mathcal{L}_{MI}$ enhances the security of $M_h$ by mutual information minimization, thereby enabling safe training of $M_h$. Meanwhile, $\mathcal{L}_D$ protects $M_t$ from model completion attacks by discouraging the model from reverting to its pretrained parameter space.

From an information-preservation perspective, since $M_{VIB}$ at the model head removes potentially sensitive semantics from $H$, it may also inadvertently erase useful prompt-related information. The adapter can compensate for this information loss by encoding the unperturbed information of $H$ and leveraging it during decoding to enrich the perturbed representation $T$ into $T'$.

# 6. Experiment Settings

To comprehensively evaluate our proposed PIDI attack and ADMI defense strategy, we designed the following experiments as detailed below. Full parameter settings and configurations are provided in Appendix B.

**Models and Dataset Settings** We conduct our experiments across 3 LLMs: **Llama3.2-3B**, **Llama3-8B**, **Qwen2.5-7B** and 3 datasets: **Dolly**(Conover et al., 2023)(General QA&Instruction following), MedicalMeadow WikiDoc PatientInfo(Han et al., 2023) (Medical QA), and Financial-qa-10k(viratt, 2024) (Finance QA). In the following sections, we will refer to MedicalMeadow WikiDoc PatientInfo as **Med** and Financial-qa-10k as **Fin**.

**Attack Settings** Apart from our proposed **PIDI**, our evaluated baseline methods include: **BiSR**(Chen et al., 2024) (SIP initialization + vanilla model inversion, VMI)[1], **Model Completion (MC)**(Fu et al., 2022; Liu et al., 2025), which leverages the target model's pretrained head to complete generated sequences, and **Vanilla Model Inversion (VMI)**(Fredrikson et al., 2015). Additionally, we assess the following ablation variants: **DSI+VMI** (our DSI initialization with vanilla inversion), **MC+VMI** (model completion initialization with vanilla inversion), and **DSI** (initialization only). Results are summarized in Table 1.

**Defense Settings** Apart from our proposed ADMI defense, we evaluate the following methods as baselines: (1) **Learning-based Defenses** design learning strategies or defense regularizers to yield robust activations against reconstruction: **DualGuard**(Liu et al., 2025) applies a local warm-up strategy with several adversarial defense regularizers to guard both the model head and tail. **MID**(Zou et al., 2023; Gu et al., 2025) uses mutual information-based regularizers to prevent data leakage from forward activations. (2) **Perturbation-based Defenses** inject noise into forward calculation to obscure private information: **DPForward**(Du et al., 2023) and **Sparsification(SP)**(Aji & Heafield, 2017; Gu et al., 2025) inject differential privacy (DP) noise or apply sparsification onto the transmitted hidden states. **SanText**(Yue et al., 2021) and **CusText**(Chen et al., 2023) employ token-wise perturbation based on an MLDP(Chatzikokolakis et al., 2013b) mechanism. Each defense is evaluated under multiple strength levels to present its overall privacy-utility trade-off as detailed in Appendix B.

**Metrics** To comprehensively evaluate our attacks and defenses, we follow the evaluation method in (Gu et al., 2025). **Main Task Performance(MP)** evaluate the model perfor-

*Table 2.* Defense Performance Against PIDI Across Fin, Med, and Dolly Datasets. Here, AP denotes $\text{AP}_{\alpha=0.5}$ and DCS denotes $\text{DCS}_{\alpha=0.5}$. Full results are reported in Tables 7 to 9.

| Defense | Llama3.2-3B | | | Llama3-8B | | | Qwen2.5-7B | | |
|---|---|---|---|---|---|---|---|---|---|
| | MP | AP | DCS | MP | AP | DCS | MP | AP | DCS |
| **Fin Dataset** | | | | | | | | | |
| w/o defense | 0.56 | 0.868 | / | 0.527 | 0.883 | / | 0.55 | 0.892 | / |
| ADMI | 0.516 | **0.017** | **0.968** | 0.515 | **0.014** | **0.987** | 0.527 | **0.004** | **0.984** |
| DualGuard | 0.459 | 0.296 | 0.819 | 0.51 | 0.446 | 0.76 | 0.542 | 0.572 | 0.712 |
| MID | 0.51 | 0.167 | 0.89 | 0.497 | 0.023 | 0.974 | 0.515 | 0.29 | 0.829 |
| DPForward | 0.549 | 0.721 | 0.662 | **0.533** | 0.85 | 0.625 | 0.533 | 0.711 | 0.665 |
| SP | **0.554** | 0.617 | 0.696 | 0.527 | 0.649 | 0.685 | **0.544** | 0.756 | 0.652 |
| SanText | 0.385 | 0.746 | 0.648 | 0.431 | 0.805 | 0.636 | 0.399 | 0.767 | 0.644 |
| CuxText | 0.507 | 0.837 | 0.628 | 0.498 | 0.849 | 0.625 | 0.464 | 0.82 | 0.632 |
| **Med Dataset** | | | | | | | | | |
| w/o defense | 0.176 | 0.901 | / | 0.187 | 0.881 | / | 0.184 | 0.801 | / |
| ADMI | 0.173 | **0.074** | 0.95 | 0.205 | 0.007 | **0.995** | 0.182 | 0.007 | **0.995** |
| DualGuard | 0.147 | 0.312 | 0.819 | 0.144 | 0.449 | 0.758 | 0.178 | 0.503 | 0.738 |
| MID | 0.169 | 0.138 | 0.911 | 0.165 | 0.027 | 0.976 | **0.191** | 0.211 | 0.87 |
| DPForward | 0.161 | 0.68 | 0.675 | 0.194 | 0.845 | 0.626 | 0.138 | 0.729 | 0.659 |
| SP | 0.162 | 0.538 | 0.724 | 0.162 | 0.672 | 0.678 | 0.168 | 0.686 | 0.673 |
| SanText | 0.141 | 0.796 | 0.64 | 0.105 | 0.9 | 0.61 | 0.149 | 0.73 | 0.659 |
| CuxText | **0.185** | 0.841 | 0.627 | 0.166 | 0.9 | 0.611 | 0.097 | 0.627 | 0.691 |
| **Dolly Dataset** | | | | | | | | | |
| w/o defense | 0.22 | 0.876 | / | 0.176 | 0.934 | / | 0.183 | 0.985 | / |
| ADMI | 0.155 | **0.011** | **0.956** | 0.205 | 0.016 | **0.989** | 0.163 | 0.27 | **0.839** |
| DualGuard | 0.146 | 0.5 | 0.737 | 0.163 | 0.511 | 0.735 | **0.171** | 0.613 | 0.698 |
| MID | 0.149 | 0.062 | 0.937 | 0.154 | 0.012 | 0.983 | **0.171** | 0.139 | 0.91 |
| DPForward | 0.16 | 0.321 | 0.813 | 0.171 | 0.608 | 0.699 | 0.162 | 0.727 | 0.66 |
| SP | 0.157 | 0.444 | 0.759 | 0.173 | 0.607 | 0.7 | **0.171** | 0.49 | 0.743 |
| SanText | 0.111 | 0.753 | 0.65 | 0.128 | 0.834 | 0.629 | 0.118 | 0.845 | 0.625 |
| CuxText | **0.167** | 0.878 | 0.617 | 0.176 | 0.937 | 0.601 | 0.133 | 0.579 | 0.709 |

mance, which is measured by the METEOR(Lavie & Agarwal, 2007) score of the models answer to the gold answer. **Attack Performance (AP).** We measure the attacks using the BLEU(Papineni et al., 2002) score between the inferred text and the ground-truth private text, defined as Attack Performance (AP). Since private data includes both the prompt input and the model-generated response, we separately define $\text{AP}_{\text{input}}$ as the AP on the input sequence and $\text{AP}_{\text{response}}$ as that on the response sequence. We aggregate them into an overall attack performance metric using a weighted formulation:

$$\text{AP}_\alpha = \alpha \cdot \text{AP}_{\text{input}} + (1 - \alpha) \cdot \text{AP}_{\text{response}} \quad (7)$$

where $\alpha \in [0, 1]$ controls the relative importance between input and response privacy. **Defense Capability Score (DCS).** To comprehensively evaluate the privacy-utility trade-off of different defenses, we adopt the Defense Capability Score (DCS) proposed in (Gu et al., 2025). Based on $\text{AP}_\alpha$, we define $\text{DCS}_{\alpha,\beta}$ as shown in Equation (8). *A higher DCS value indicates a more favorable balance between privacy protection and model utility.* Using larger $\alpha$ indicates placing greater emphasis on protecting input privacy over response privacy, while larger values of $\beta$ indicate a stronger preference for model utility over privacy. Unless otherwise specified, we set $\alpha = 0.5, \beta = 0.5$ throughout this paper.

$$\text{DCS}_{\alpha,\beta} = \frac{1}{1 + \sqrt{(1 - \beta)(\text{AP}_\alpha - \text{AP}_\alpha^*)^2 + \beta(\text{MP} - \text{MP}^*)^2}} \quad (8)$$

---

[1] Since gradient is not available in open-end generation task, the gradient matching module in the original BiSR is not implemented

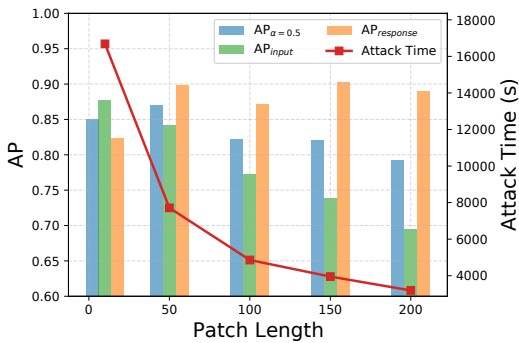

*Figure 6.* Changing trend of AP and Attack time with respect to Patch Length $l$ (Evaluated with Llama3.2-3B on Fin Dataset)

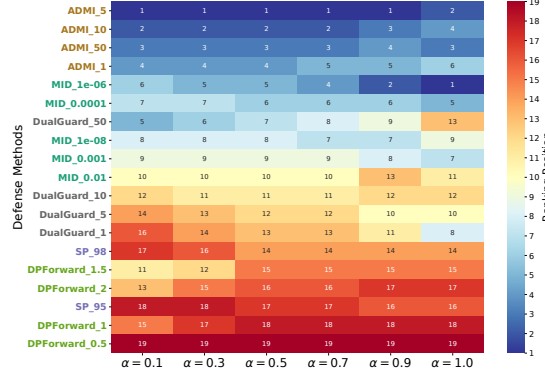

*Figure 7.* DCS ranking with Different $\alpha$ values (Evaluated with Llama3-3B on Fin Dataset)

## 7. Results and Analysis

### 7.1. Attack Performance Analysis

As shown in Table 1, our proposed PIDI attack generally achieves the highest AP across various tasks and datasets, demonstrating strong attack effectiveness. The DSI initialization in DSI+VMI leverages both input and response features, yielding superior performance over other initializations such as MC+VMI and BiSR (SIP+VMI). Furthermore, as shown in Figure 4, traditional VMI-based methods(e.g. DSI+VMI, BiSR) suffer significant AP degradation as sequence length increases, whereas our patched inversion technique PMI maintains stable and high AP across varying lengths while its marginal benefit diminishes for short sequences in smaller models These results validate the effectiveness of both our DSI and PMI designs in PIDI.

We also present an ablation study on the impact of patch length $l$, as shown in Figure 6. As $l$ increases, $AP_{input}$ degrades significantly, while $AP_{response}$ remains largely stable. Meanwhile, a smaller $l$ leads to substantially longer attack times required. Considering the trade-off between reconstruction quality and attack efficiency, we select $l = 50$ in our experiments.

### 7.2. Defense Performance Analysis

Considering the MP-AP trend depicted in Figure 5 and Figure 8, the performance dots of our proposed ADMI defense lie significantly towards the lower-right region of the figure, indicating a superior MP–AP trade-off. Notably, the adaptive weighting strategy allows ADMI to increase its defense strength while incurring minimal MP degradation, demonstrating its effectiveness in balancing privacy protection and utility retention.

Results in Table 2 further demonstrate that ADMI achieves minimal MP loss while maintaining the lowest AP among all baselines across various models and tasks. Its optimal DCS score confirms its overall effectiveness, highlighting a favorable privacy–utility trade-off.

As shown in Figure 8, DualGuard and MID perform com-

parably to ADMI under the BiSR attack, which targets only the model head. However, they fail to maintain robust performance under PIDI and MC, which additionally attack from the model tail. This observation highlights that ADMI's adapter-based design confers a pronounced advantage in protecting the model tail. In comparison to these learning-based defenses, perturbation-based methods generally exhibit weaker performance. SP achieves slightly better results than DPForward, whereas token-wise perturbation approaches, such as SanText and CusText, yield the weakest results. This indicates that, in open-ended generation, simply perturbing input tokens at the model head is insufficient for comprehensive protection, as such strategies cannot mitigate attacks targeting the model tail.

To evaluate the overall defense performance fairly, we compute $DCS_{\alpha,\beta}$ (Equation (8)) with $\beta = 0.5$ and varying $\alpha$ values. Higher $\alpha$ indicates attaching more importance to input privacy versus the response privacy. The resulting ranks are visualized in Figures 7, 9 and 10, where lower numbers (bluer cells) indicate better performance. ADMI consistently ranks among the highest, confirming that it consistently provides favorable defense performance across various preferences.

For completeness, we present further ablation studies on both attack and defense settings, including split configurations and auxiliary data size, along with additional experimental results in Appendix C.

## 8. Conclusions and Limitations

In this work, we investigate the dual-sided privacy risks inherent in Split-LLMs by introducing a novel attack, **P**atched Model **I**nversion with **D**ual-Sided **I**nitialization (**PIDI**), addressing the privacy of both the input prompts and model-generated responses. By effectively exploiting information leakage from both model ends and leveraging a patched inversion strategy, PIDI achieves substantially stronger attack performance than existing methods. To counter this threat, we further propose the **A**dapter-based **D**ualGuard

with **M**utual **I**nformation Defense(**ADMI**), a dual-sided defense framework integrated with a novel adapter-based strategy. This formulation provides robust protection for both model ends while incurring minimal utility loss, thereby enabling a more favorable privacy–utility trade-off.

Despite these promising results, our work has several limitations. ADMI introduces additional computational overhead, which may limit its applicability in latency-sensitive or large-scale deployment scenarios. Moreover, while PMI is particularly effective in long-sequence attack settings, its advantage becomes less pronounced on short texts or smaller models, and it may incur higher attack latency.

## Impact Statement

This work advances the understanding of privacy risks and defenses in Split-LLM. By identifying and formalizing dual-sided data leakage from both input prompts and generative responses, our findings can help practitioners design more robust systems and make informed deployment decisions.

At the same time, our proposed PIDI attack methodology may be misused to extract sensitive information from inadequately protected Split-LLM systems. We mitigate this risk by presenting an effective defense framework ADMI, which substantially reduces vulnerability while preserving model utility, and by releasing our code to support reproducibility and responsible evaluation of privacy-preserving techniques.

We believe this work contributes positively by promoting transparency around emerging privacy threats and by providing practical mechanisms to strengthen safeguards for real-world Split-LLM deployments.

## Acknowledgements

This work was supported by the Presidential Young Scholar Scheme(Project No. P0056638)RIFL(Project No. 4-CG00), and the RIAIoT(Project No. P0059914) at The Hong Kong Polytechnic University.

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

# A. Detailed Derivation of the Regularizer $L_{\text{MI}}$

We first show that the mutual information term $I(E; H')$ establishes a theoretical upper bound on information leakage through intermediate activations. Since $T$ is deterministic given $H'$, we have $I(X; H', T) = I(X; H') + I(X; T|H') = I(X; H')$. Let $\hat{X} = \mathcal{A}(H', T)$ be any attacker-reconstructed input (deterministic or randomized). By data processing inequality, $I(X; \hat{X}) \leq I(X; H', T) = I(X; H') \leq L_{MI}$. Thus, for any attacker reconstructed result $\hat{X} = \mathcal{A}(H', T)$, $I(X; \hat{X}) \leq I(X; H', T) \leq L_{MI}$.

However, directly optimizing $I(E; H')$ is intractable. To obtain a tractable training objective, we introduce a variational information bottleneck (VIB) model $M_{\text{VIB}}$ between $M_h$ and $M_b$. This VIB model acts as a stochastic bottleneck and it takes an encoder-decoder structure: its encoder projects $H$ to Gaussian parameters $(\boldsymbol{\mu}, \boldsymbol{\sigma})$ and a latent variable $Z$ is sampled as $Z = \boldsymbol{\mu} + \boldsymbol{\sigma} \odot \boldsymbol{\epsilon}$, $\boldsymbol{\epsilon} \sim \mathcal{N}(0, I)$. The decoder then maps $Z$ to a perturbed representation $H'$.

Given the Markov chain $E \rightarrow H \rightarrow Z \rightarrow H'$, the Data Processing Inequality ensures $I(E, H') \leq I(H, Z)$.

$$I(E, H') \leq I(E, Z) \leq I(H, Z) \tag{9}$$

We can further derive the expected KL divergence between the variational posterior $q_\phi(Z|H)$ and a chosen prior $p(z) = \mathcal{N}(0, I)$ as a variational upper bound surrogate of $I(H, Z)$:

$$
\begin{aligned}
I(H; Z) &= \mathbb{E}_{p(H)}\left[\text{KL}\left(q_\phi(z|H) \,\|\, q_\phi(z)\right)\right] = \mathbb{E}_{p(H)}\left[\text{KL}\left(q_\phi(Z|H) \,\|\, p(Z)\right)\right] - \text{KL}\left(q_\phi(Z) \,\|\, p(Z)\right) \\
&\leq \mathbb{E}_{p(H)}\left[\text{KL}\left(q_\phi(Z|H) \,\|\, p(Z)\right)\right]
\end{aligned}
\tag{10}
$$

where $\phi$ denotes the parameter of the VIB encoder and $q_\phi(z) = \int q_\phi(z|H)p(H)dH$ is the marginal distribution of $Z$, also known as the aggregated posterior. Therefore we can further derive the upper bound of $I(E, H')$ as:

$$
\begin{aligned}
\mathcal{L}_{\text{MI}} &= \text{UpperBound}(I(E, H')) = \mathbb{E}_{H \sim p(H)}\left[\text{KL}\left(q_\phi(\mathbf{z}|H) \,\|\, p(\mathbf{z})\right)\right] \\
&\approx \frac{1}{N}\sum_{i=1}^{N}\text{KL}\left(\mathcal{N}(\boldsymbol{\mu}_i, \boldsymbol{\sigma}_i^2) \,\|\, \mathcal{N}(0, I)\right) \\
&= \frac{1}{N}\sum_{i=1}^{N}\frac{1}{2}\sum_{j=1}^{d}\left(\mu_{ij}^2 + \sigma_{ij}^2 - \log(\sigma_{ij}^2) - 1\right)
\end{aligned}
\tag{11}
$$

# B. Detailed Experiment Settings

**Model and Dataset Configurations**  For all models, we set the partition settings as $n_h = 4$ and $n_t = 4$, leaving $n_b = 20, 20, 24$ for Llama3.2-3B/Qwen2.5-7B/Llama3-8B, respectively. The Dolly(Conover et al., 2023) dataset is an open-source collection of instruction-following records, spanning multiple behavioral categories. The MedicalMeadow WikiDoc PatientInfo dataset(Med) is the Wikidoc Patient Information subset of the training set in (Han et al., 2023). It is a medical QA dataset with medical question-answer pairs extracted from WikiDoc, a collaborative platform for medical professionals to share and contribute to up-to-date medical knowledge. The financial-QA-10k dataset(Fin)(virattt, 2024) is a collection of 10,000 question-answer pairs derived from corporate 10-K filings, covering diverse topics in financial analysis, company operations, and corporate strategy. For all datasets, we split 5% of the full set as the test set and the remaining part as train set.

**Training Configurations**  We set learning rate to 2e-4 and batch size to 8 throughout the work. Convergence is marked with an early-stop strategy. Notably, the learning rate schedule is critical for the training stability of our ADMI defense as detailed in Table 4. During local warm-up stage, the randomly initialized adapters are trained with a larger learning rate to quickly align with the pretrained backbone, while $M_{h/t}$ uses a smaller learning rate to induce a smooth parameter shift and prevent drastic representation drift. During full training, we increase the backbone learning rate to enable deeper task adaptation, and train the newly introduced $M_{\text{VIB}}$ with a larger learning rate for fast convergence and effective integration.

**Defense Settings**  In Figures 5 and 8, we evaluate each defense under a range of increasing defense strengths. The corresponding controlled hyper-parameter values, ordered from weakest to strongest, are summarized in Table 3. Parameters

Table 3. Summary of defenses hyper-parameters.

| Defense | Controlled Hyper-parameter Values |
|---|---|
| ADMI | $\beta = 1, 5, \mathbf{10}, 50$ |
| DualGuard | $\lambda_3 = 1, 5, \mathbf{10}, 50$ |
| MID | $\lambda = 10^{-6}, \mathbf{10^{-4}}, 10^{-3}, 0.01$ |
| TO | $\epsilon = 10, \mathbf{5}, 2, 1$ |
| DPForward | noise scale$= 0.5, \mathbf{1}, 1.5, 2$ |
| SP | $r = 90.0\%, \mathbf{95.0\%}, 98.0\%, 99.5\%$ |
| SanText | $p = 0.02, \mathbf{0.05}, 0.08, 0.1$ |
| CusText | $\epsilon = 150, \mathbf{100}, 50, 30$ |

Table 4. Learning rate configurations for different model components in PIDI.

| Model | Stage | Learning Rate | | | | |
|---|---|---|---|---|---|---|
| | | $M_{\mathrm{h}}$ | $M_{\mathrm{t}}$ | $M_{\mathrm{h/t}}$ | $A_e/A_d$ | $M_{\mathrm{VIB}}$ |
| Llama3.2-3B | Local-Warmup | 2e-5 | / | 2e-5 | 4e-4 | / |
| | Full Training | 2e-4 | 2e-4 | 2e-4 | 2e-4 | 4e-4 |
| Qwen2.5-7B | Local-Warmup | 2e-5 | / | 2e-4 | 2e-4 | / |
| | Full Training | 2e-4 | 4e-4 | 2e-4 | 2e-4 | 4e-4 |
| Llama3-8B | Local-Warmup | 2e-5 | / | 2e-4 | 2e-4 | / |
| | Full Training | 2e-4 | 4e-4 | 2e-4 | 2e-4 | 4e-4 |

used in Tables 2 and 7 to 9 are highlighted in bold. For ADMI, defense strength is controlled via the model distance regularizer weight $\beta$. In DualGuard, we fix $\lambda_1 = 30$ and $\lambda_2 = 70$, and vary $\lambda_3$ (the anti-pretrained-tail loss weight, analogous to ADMI's model distance regularizer) to adjust defense strength. MID uses the MI regularizer weight $\lambda$ to tune defense strength. TO sets $w_{\mathrm{away}} = 0.5$, $w_{\mathrm{close}} = 0.1$, and $n_{\mathrm{cluster}} = 100$, and varies $\epsilon$ to control defense strength. DPForward adjusts the noise scale injected into the embedding space. SP controls defense via the sparsification rate $r$, i.e., the fraction of sparsified tensor coordinates. SanText adjusts the fraction $p$ of perturbed sensitive words sanitized, while CusText fixes top-$k = 100$ and varies $\epsilon$ for DP noise. For both ADMI and MID, the inserted $M_{\mathrm{VIB}}$ encoder and decoder use a single-layer MLP with ReLU. For both ADMI and DualGuard, we set the local warmup epoch to 2.

**Attack Settings** We randomly select 200 samples from the test dataset to evaluate attack performance, on which the Split-LLM system performs open-end generation. For all vanilla model inversion training in our experiments (BiSR/VMI), we employ an early-stop strategy to determine convergence, with a maximum of 1000 inversion steps. For PIDI, each patched inversion also uses early stopping with a smaller maximum of 300 steps, while the final full inversion is limited to 500 steps. The inversion learning rate is set to 0.001 for all methods. The auxiliary dataset for SIP training consists of 50 samples randomly drawn from the training set.

Table 5. Attack Performance($\mathrm{AP_{input}}$) of the evaluated attack methods

| | Fin | | | Med | | | Dolly | | |
|---|---|---|---|---|---|---|---|---|---|
| | Llama3.2-3B | Llama3-8B | Qwen2.5-7B | Llama3.2-3B | Llama3-8B | Qwen2.5-7B | Llama3.2-3B | Llama3-8B | Qwen2.5-7B |
| PIDI(DSI+PMI) | **0.844** | **0.873** | **0.877** | **0.951** | **0.931** | **0.945** | **0.857** | **0.953** | **0.996** |
| DSI+VMI | 0.753 | 0.579 | 0.678 | 0.94 | 0.893 | 0.93 | 0.699 | 0.67 | 0.757 |
| MC+VMI | 0 | 0.009 | 0.022 | 0 | 0.004 | 0 | 0.002 | 0.083 | 0.005 |
| BiSR(SIP+VMI) | 0.75 | 0.726 | 0.678 | 0.937 | 0.893 | 0.93 | 0.673 | 0.603 | 0.789 |
| DSI | 0.582 | 0.557 | 0.678 | 0.902 | 0.893 | 0.921 | 0.622 | 0.6 | 0.673 |
| SIP | 0.582 | 0.557 | 0.678 | 0.902 | 0.893 | 0.921 | 0.623 | 0.6 | 0.673 |
| MC | 0 | 0 | 0.022 | 0 | 0 | 0 | 0 | 0 | 0.004 |
| VMI | 0.005 | 0.27 | 0.035 | 0.005 | 0.214 | 0.004 | 0.006 | 0.002 | 0.002 |

Table 6. Attack Performance($\mathrm{AP_{response}}$) of the evaluated attack methods

| | Fin | | | Med | | | Dolly | | |
|---|---|---|---|---|---|---|---|---|---|
| | Llama3.2-3B | Llama3-8B | Qwen2.5-7B | Llama3.2-3B | Llama3-8B | Qwen2.5-7B | Llama3.2-3B | Llama3-8B | Qwen2.5-7B |
| PIDI(DSI+PMI) | 0.891 | **0.893** | **0.907** | 0.851 | **0.831** | 0.658 | 0.895 | **0.916** | **0.975** |
| DSI+VMI | **0.904** | 0.854 | 0.863 | **0.911** | 0.522 | 0.551 | 0.903 | 0.881 | 0.87 |
| MC+VMI | 0.86 | 0.836 | 0.863 | 0.746 | 0.525 | **0.847** | **0.944** | 0.825 | 0.819 |
| BiSR(SIP+VMI) | 0.58 | 0.495 | 0.414 | 0.764 | 0.198 | 0.482 | 0.292 | 0.043 | 0.544 |
| DSI | 0.8 | 0.836 | 0.863 | 0.471 | 0.522 | 0.49 | 0.792 | 0.783 | 0.807 |
| SIP | 0.199 | 0.138 | 0.414 | 0.259 | 0.181 | 0.412 | 0.06 | 0.014 | 0.227 |
| MC | 0.8 | 0.836 | 0.863 | 0.471 | 0.522 | 0.49 | 0.646 | 0.783 | 0.807 |
| VMI | 0.008 | 0.354 | 0.414 | 0.006 | 0.339 | 0.103 | 0.003 | 0.002 | 0.055 |

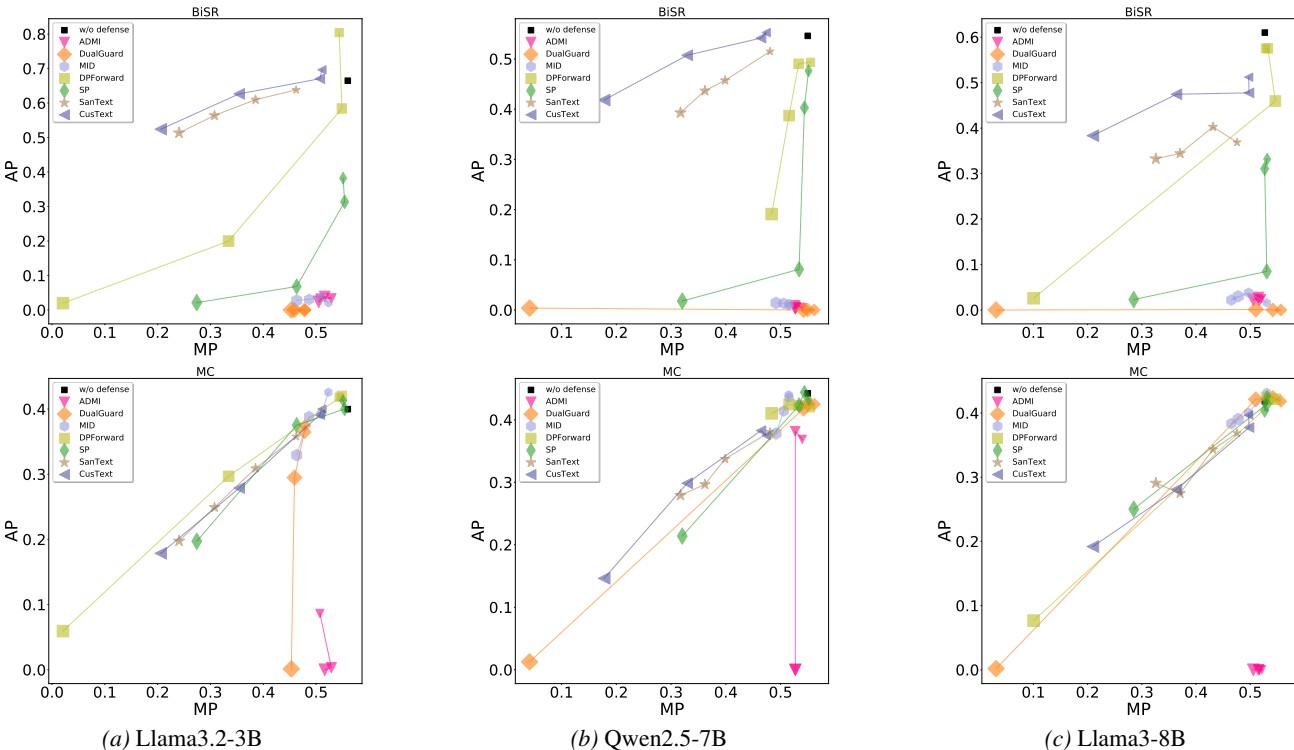

*Figure 8.* Defense Performance(MP-AP) for defending BiSR/MC attack on the Fin Dataset. Dot size represents the defense strength. Here AP denotes $AP_{\alpha=0.5}$.

*Table 7.* Full Defense Performance Against PIDI attack Across Fin, Med, and Dolly Datasets.

| Defense | Llama3.2-3B | | | Llama3-8B | | | Qwen2.5-7B | | |
|---|---|---|---|---|---|---|---|---|---|
| | MP | $AP_{response}$ | $AP_{input}$ | MP | $AP_{response}$ | $AP_{input}$ | MP | $AP_{response}$ | $AP_{input}$ |
| **Fin Dataset** | | | | | | | | | |
| w/o defense | 0.56 | 0.891 | 0.844 | 0.527 | 0.893 | 0.873 | 0.55 | 0.907 | 0.877 |
| ADMI | 0.516 | 0.009 | 0.025 | 0.515 | 0.017 | 0.012 | 0.527 | 0.003 | 0.005 |
| DualGuard | 0.459 | 0.592 | 0 | 0.51 | 0.86 | 0.032 | 0.542 | 0.906 | 0.238 |
| MID | 0.51 | 0.308 | 0.026 | 0.497 | 0.033 | 0.013 | 0.515 | 0.571 | 0.01 |
| DPForward | 0.549 | 0.78 | 0.662 | 0.533 | 0.888 | 0.812 | 0.533 | 0.855 | 0.568 |
| SP | 0.554 | 0.763 | 0.471 | 0.527 | 0.748 | 0.55 | 0.544 | 0.904 | 0.607 |
| SanText | 0.385 | 0.693 | 0.8 | 0.431 | 0.754 | 0.856 | 0.399 | 0.705 | 0.83 |
| CuxText | 0.507 | 0.832 | 0.842 | 0.498 | 0.833 | 0.864 | 0.464 | 0.772 | 0.867 |
| **Med Dataset** | | | | | | | | | |
| w/o defense | 0.176 | 0.851 | 0.951 | 0.187 | 0.831 | 0.931 | 0.184 | 0.658 | 0.945 |
| ADMI | 0.173 | 0.124 | 0.024 | 0.205 | 0.006 | 0.007 | 0.182 | 0.001 | 0.012 |
| DualGuard | 0.147 | 0.623 | 0.001 | 0.144 | 0.598 | 0.3 | 0.178 | 0.873 | 0.133 |
| MID | 0.169 | 0.229 | 0.046 | 0.165 | 0.044 | 0.011 | 0.191 | 0.411 | 0.011 |
| DPForward | 0.161 | 0.506 | 0.854 | 0.194 | 0.844 | 0.846 | 0.138 | 0.585 | 0.874 |
| SP | 0.162 | 0.401 | 0.676 | 0.162 | 0.654 | 0.69 | 0.168 | 0.572 | 0.8 |
| SanText | 0.141 | 0.695 | 0.896 | 0.105 | 0.891 | 0.91 | 0.149 | 0.562 | 0.897 |
| CuxText | 0.185 | 0.746 | 0.935 | 0.166 | 0.841 | 0.958 | 0.097 | 0.302 | 0.952 |
| **Dolly Dataset** | | | | | | | | | |
| w/o defense | 0.22 | 0.895 | 0.857 | 0.176 | 0.916 | 0.953 | 0.183 | 0.975 | 0.996 |
| ADMI | 0.155 | 0.002 | 0.02 | 0.205 | 0.015 | 0.016 | 0.163 | 0.489 | 0.051 |
| DualGuard | 0.146 | 0.986 | 0.013 | 0.163 | 0.986 | 0.036 | 0.171 | 0.938 | 0.288 |
| MID | 0.149 | 0.099 | 0.026 | 0.154 | 0.011 | 0.012 | 0.171 | 0.272 | 0.005 |
| DPForward | 0.16 | 0.256 | 0.385 | 0.171 | 0.651 | 0.565 | 0.162 | 0.857 | 0.598 |
| SP | 0.157 | 0.46 | 0.429 | 0.173 | 0.63 | 0.583 | 0.171 | 0.662 | 0.318 |
| SanText | 0.111 | 0.728 | 0.779 | 0.128 | 0.763 | 0.905 | 0.118 | 0.761 | 0.928 |
| CuxText | 0.167 | 0.874 | 0.882 | 0.176 | 0.904 | 0.97 | 0.133 | 0.33 | 0.827 |

*Table 8.* Full Defense Performance Against BiSR attack Across Fin, Med, and Dolly Datasets.

| Defense | Llama3.2-3B | | | Llama3-8B | | | Qwen2.5-7B | | |
|---|---|---|---|---|---|---|---|---|---|
| | MP | $AP_{response}$ | $AP_{input}$ | MP | $AP_{response}$ | $AP_{input}$ | MP | $AP_{response}$ | $AP_{input}$ |
| **Fin Dataset** | | | | | | | | | |
| w/o defense | 0.56 | 0.58 | 0.75 | 0.527 | 0.495 | 0.726 | 0.55 | 0.414 | 0.678 |
| ADMI | 0.516 | 0.036 | 0.042 | 0.515 | 0.035 | 0.02 | 0.527 | 0.002 | 0.005 |
| DualGuard | 0.459 | 0 | 0 | 0.51 | 0.002 | 0 | 0.542 | 0 | 0 |
| MID | 0.51 | 0.038 | 0.036 | 0.497 | 0.048 | 0.028 | 0.515 | 0.012 | 0.012 |
| DPForward | 0.549 | 0.489 | 0.68 | 0.533 | 0.508 | 0.643 | 0.533 | 0.413 | 0.568 |
| SP | 0.554 | 0.166 | 0.462 | 0.527 | 0.191 | 0.43 | 0.544 | 0.304 | 0.502 |
| SanText | 0.385 | 0.49 | 0.73 | 0.431 | 0.25 | 0.556 | 0.399 | 0.316 | 0.599 |
| CuxText | 0.507 | 0.56 | 0.781 | 0.498 | 0.315 | 0.64 | 0.464 | 0.382 | 0.701 |
| **Med Dataset** | | | | | | | | | |
| w/o defense | 0.176 | 0.764 | 0.937 | 0.187 | 0.198 | 0.893 | 0.184 | 0.482 | 0.93 |
| ADMI | 0.173 | 0.023 | 0.074 | 0.205 | 0.011 | 0.001 | 0.182 | 0.013 | 0.011 |
| DualGuard | 0.147 | 0 | 0 | 0.144 | 0.004 | 0 | 0.178 | 0 | 0 |
| MID | 0.169 | 0.02 | 0.047 | 0.165 | 0.026 | 0.023 | 0.191 | 0.012 | 0.012 |
| DPForward | 0.161 | 0.314 | 0.865 | 0.194 | 0.656 | 0.858 | 0.138 | 0.364 | 0.876 |
| SP | 0.162 | 0.139 | 0.675 | 0.162 | 0.204 | 0.801 | 0.168 | 0.402 | 0.799 |
| SanText | 0.141 | 0.706 | 0.911 | 0.105 | 0.248 | 0.785 | 0.149 | 0.464 | 0.854 |
| CuxText | 0.185 | 0.846 | 0.947 | 0.166 | 0.614 | 0.906 | 0.097 | 0.361 | 0.949 |
| **Dolly Dataset** | | | | | | | | | |
| w/o defense | 0.22 | 0.292 | 0.673 | 0.176 | 0.043 | 0.603 | 0.183 | 0.544 | 0.789 |
| ADMI | 0.155 | 0.015 | 0.036 | 0.205 | 0.038 | 0.017 | 0.163 | 0.017 | 0.052 |
| DualGuard | 0.146 | 0 | 0 | 0.163 | 0 | 0 | 0.171 | 0 | 0 |
| MID | 0.149 | 0.015 | 0.029 | 0.154 | 0.01 | 0.024 | 0.171 | 0 | 0 |
| DPForward | 0.16 | 0.129 | 0.608 | 0.171 | 0.053 | 0.601 | 0.162 | 0.341 | 0.648 |
| SP | 0.157 | 0.032 | 0.441 | 0.173 | 0.013 | 0.497 | 0.171 | 0.146 | 0.328 |
| SanText | 0.111 | 0.22 | 0.598 | 0.128 | 0.046 | 0.548 | 0.118 | 0.487 | 0.777 |
| CuxText | 0.167 | 0.337 | 0.663 | 0.176 | 0.146 | 0.608 | 0.133 | 0.214 | 0.786 |

*Table 9.* Full Defense Performance Against Model Completion(MC) Attack Across Fin, Med, and Dolly Datasets.

| Defense | Llama3.2-3B | | | Llama3-8B | | | Qwen2.5-7B | | |
|---|---|---|---|---|---|---|---|---|---|
| | MP | $AP_{response}$ | $AP_{input}$ | MP | $AP_{response}$ | $AP_{input}$ | MP | $AP_{response}$ | $AP_{input}$ |
| **Fin Dataset** | | | | | | | | | |
| w/o defense | 0.56 | 0.8 | 0 | 0.527 | 0.836 | 0 | 0.55 | 0.863 | 0.022 |
| ADMI | 0.516 | 0 | 0 | 0.515 | 0 | 0 | 0.527 | 0 | 0 |
| DualGuard | 0.459 | 0.589 | 0 | 0.51 | 0.839 | 0.003 | 0.542 | 0.836 | 0 |
| MID | 0.51 | 0.785 | 0 | 0.497 | 0.8 | 0 | 0.515 | 0.871 | 0 |
| DPForward | 0.549 | 0.84 | 0 | 0.533 | 0.844 | 0 | 0.533 | 0.846 | 0 |
| SP | 0.554 | 0.8 | 0 | 0.527 | 0.808 | 0 | 0.544 | 0.888 | 0 |
| SanText | 0.385 | 0.615 | 0.003 | 0.431 | 0.686 | 0 | 0.399 | 0.649 | 0.026 |
| CuxText | 0.507 | 0.75 | 0.032 | 0.498 | 0.754 | 0.001 | 0.464 | 0.728 | 0.037 |
| **Med Dataset** | | | | | | | | | |
| w/o defense | 0.176 | 0.471 | 0 | 0.187 | 0.522 | 0 | 0.184 | 0.49 | 0 |
| ADMI | 0.173 | 0.348 | 0.004 | 0.205 | 0 | 0 | 0.182 | 0 | 0 |
| DualGuard | 0.147 | 0.37 | 0.002 | 0.144 | 0.317 | 0.008 | 0.178 | 0.485 | 0.007 |
| MID | 0.169 | 0.376 | 0 | 0.165 | 0.431 | 0.006 | 0.191 | 0.561 | 0 |
| DPForward | 0.161 | 0.415 | 0 | 0.194 | 0.528 | 0 | 0.138 | 0.423 | 0 |
| SP | 0.162 | 0.399 | 0 | 0.162 | 0.474 | 0 | 0.168 | 0.462 | 0 |
| SanText | 0.141 | 0.416 | 0.001 | 0.105 | 0.758 | 0.109 | 0.149 | 0.39 | 0.002 |
| CuxText | 0.185 | 0.441 | 0.001 | 0.166 | 0.514 | 0 | 0.097 | 0.249 | 0 |
| **Dolly Dataset** | | | | | | | | | |
| w/o defense | 0.22 | 0.646 | 0 | 0.176 | 0.783 | 0 | 0.183 | 0.807 | 0.004 |
| ADMI | 0.155 | 0.001 | 0.005 | 0.205 | 0 | 0 | 0.163 | 0.757 | 0 |
| DualGuard | 0.146 | 0.805 | 0.01 | 0.163 | 0.758 | 0.007 | 0.171 | 0.81 | 0.008 |
| MID | 0.149 | 0.726 | 0 | 0.154 | 0.656 | 0.004 | 0.171 | 0.733 | 0 |
| DPForward | 0.16 | 0.814 | 0.001 | 0.171 | 0.704 | 0 | 0.162 | 0.844 | 0.002 |
| SP | 0.157 | 0.767 | 0.003 | 0.173 | 0.779 | 0 | 0.171 | 0.82 | 0.021 |
| SanText | 0.111 | 0.652 | 0.003 | 0.128 | 0.657 | 0 | 0.118 | 0.632 | 0.003 |
| CuxText | 0.167 | 0.754 | 0.042 | 0.176 | 0.768 | 0 | 0.133 | 0.308 | 0.062 |

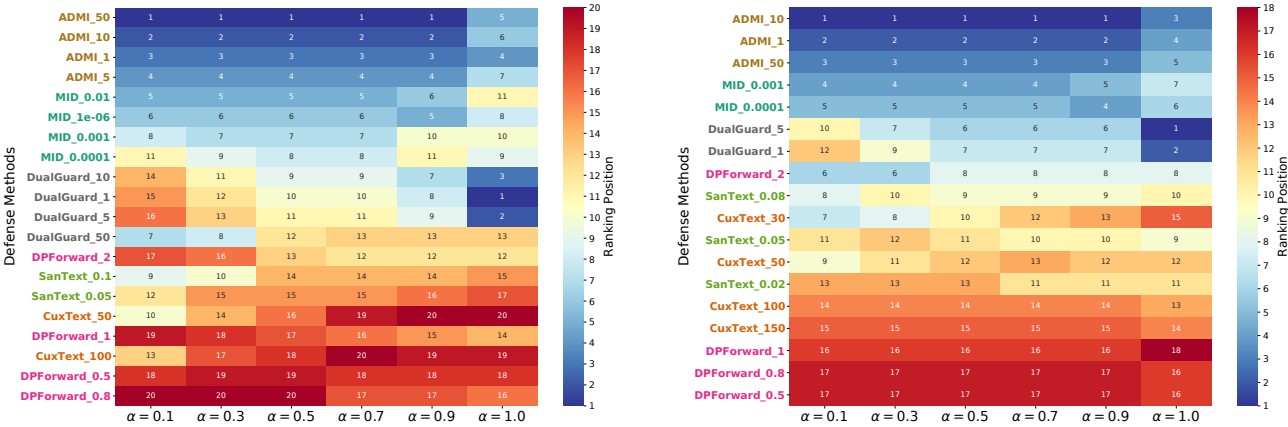

*Figure 9.* DCS ranking with Different $\alpha$ values (Evaluated with Qwen2.5-7B on Fin Dataset)

*Figure 10.* DCS ranking with Different $\alpha$ values (Evaluated with Llama3-8B on Fin Dataset)

*Table 10.* Split point ablation on Llama3.2-3B, Fin Dataset, under PIDI attack.

| model | defense | $n_h$ | $n_t$ | $MP$ | $AP_{\alpha=0.5}$ | $AP_{response}$ | $AP_{input}$ |
|---|---|---|---|---|---|---|---|
| Llama3.2-3B | ADMI | 3 | 3 | 0.474 | 0.009 | 0.009 | 0.010 |
| Llama3.2-3B | ADMI | 4 | 4 | 0.516 | 0.017 | 0.009 | 0.025 |
| Llama3.2-3B | ADMI | 5 | 5 | 0.505 | 0.004 | 0.001 | 0.006 |
| Llama3.2-3B | ADMI | 6 | 6 | 0.496 | 0.010 | 0.005 | 0.014 |
| Llama3.2-3B | wo | 3 | 3 | 0.560 | 0.950 | 0.946 | 0.954 |
| Llama3.2-3B | wo | 4 | 4 | 0.560 | 0.868 | 0.891 | 0.844 |
| Llama3.2-3B | wo | 5 | 5 | 0.560 | 0.784 | 0.842 | 0.726 |
| Llama3.2-3B | wo | 6 | 6 | 0.560 | 0.726 | 0.828 | 0.623 |

# C. Further Ablations

## C.1. Ablations on Split Point

To further assess the impact of split configurations in Split-LLM systems, we additionally evaluate different split points with $n_h = n_t \in \{3, 4, 5, 6\}$ on Llama3.2-3B over the Fin dataset under the PIDI attack. The results are reported in Table 10.

Overall, the attack performance without defense gradually decreases as the split point moves deeper into the modelsince larger model head and tail are harder to invert. In contrast, ADMI consistently maintains strong protection performance across all split settings, reducing $AP$ to near-zero values while preserving stable utility.

We also observe that, although deeper split points naturally lower $AP$ even without defense, the remaining leakage is still substantial (e.g., $AP_{\alpha=0.5} = 0.726$ at the 6-6 split). This result further highlights the necessity of dedicated privacy-preserving mechanisms such as ADMI in practical Split-LLM deployments.

## C.2. Ablations on auxiliary dataset

To evaluate the dependence of PIDI on attacker-side auxiliary data, we conduct additional ablation studies by varying the auxiliary dataset size used for SIP initialization as shown in Figure 11.

We observe that reducing the auxiliary dataset size degrades the quality of the initial SIP reconstruction, especially in extremely low-resource settings. However, the final $AP$ of PIDI remains relatively stable once a small amount of auxiliary data is available. This result suggests that the subsequent PMI refinement stage can effectively compensate for imperfect initialization. These findings demonstrate that PIDI is less sensitive to auxiliary data scarcity compared to SIP-only attacks, highlighting the synergistic interaction between initialization and iterative refinement in our framework.

## C.3. Computation overhead analysis

We further analyze the computational overhead introduced by ADMI on the Dolly dataset(see Table 11).

*Table 11.* Computation overhead analysis on the Dolly dataset.

| Model | Method | Inference Throughput (token/s/GPU) | Training Speed (s/iter) | Warmup Training Speed (s/iter) | Total Training Time (s) |
|---|---|---|---|---|---|
| Qwen2.5-7B | w/o defense | 24.892 | 0.915 | / | 13888.013 |
| Qwen2.5-7B | ADMI | 20.418 | 1.381 | 0.449 | 21866.425 |
| Llama3-8B | w/o defense | 23.354 | 0.996 | / | 17182.315 |
| Llama3-8B | ADMI | 17.486 | 1.449 | 0.482 | 21843.745 |

Overall, ADMI introduces moderate computational overhead due to the additional MID optimization and adapter-based warm-up stage. Compared with the undefended setting, inference throughput decreases by approximately 18% on Qwen2.5-7B and 25% on Llama3-8B. Training time also increases because of the additional optimization objectives and local warm-up process.

We therefore consider computational efficiency an important limitation of the current design and a promising direction for future work, particularly in scenarios where both strong privacy protection and low-latency deployment are required.

*Table 12.* Additional privacy leakage metrics ($AP_{\alpha=0.5}$).

| model | dataset | method | EntityRecovery-F1 | BERTScore-F1 | BLEURT | BLEU(Origin) |
|---|---|---|---|---|---|---|
| Llama3.2-3B | Fin | w/o defense | 0.868 | 0.977 | 0.739 | 0.868 |
| Llama3.2-3B | Fin | ADMI | **0.057** | **0.753** | **0.148** | **0.017** |
| Llama3.2-3B | Fin | DualGuard | 0.393 | 0.864 | 0.447 | 0.296 |
| Llama3.2-3B | Fin | MID | 0.236 | 0.807 | 0.299 | 0.167 |
| Llama3.2-3B | Fin | random baseline | 0.003 | 0.746 | 0.109 | 0.000 |
| Llama3.2-3B | Med | w/o defense | 0.771 | 0.981 | 0.797 | 0.901 |
| Llama3.2-3B | Med | ADMI | **0.057** | **0.749** | **0.149** | **0.074** |
| Llama3.2-3B | Med | DualGuard | 0.397 | 0.873 | 0.406 | 0.312 |
| Llama3.2-3B | Med | MID | 0.250 | 0.815 | 0.277 | 0.138 |
| Llama3.2-3B | Med | random baseline | 0.079 | 0.748 | 0.103 | 0.000 |
| Qwen2.5-7B | Fin | w/o defense | 0.839 | 0.980 | 0.703 | 0.892 |
| Qwen2.5-7B | Fin | ADMI | **0.000** | **0.743** | **0.127** | **0.00**4 |
| Qwen2.5-7B | Fin | DualGuard | 0.629 | 0.925 | 0.635 | 0.572 |
| Qwen2.5-7B | Fin | MID | 0.340 | 0.840 | 0.382 | 0.290 |
| Qwen2.5-7B | Fin | random baseline | 0.002 | 0.741 | 0.104 | 0.000 |
| Qwen2.5-7B | Med | w/o defense | 0.674 | 0.976 | 0.711 | 0.801 |
| Qwen2.5-7B | Med | ADMI | **0.056** | **0.733** | **0.126** | **0.007** |
| Qwen2.5-7B | Med | DualGuard | 0.688 | 0.923 | 0.575 | 0.503 |
| Qwen2.5-7B | Med | MID | 0.309 | 0.827 | 0.275 | 0.211 |
| Qwen2.5-7B | Med | random baseline | 0.081 | 0.743 | 0.099 | 0.000 |

## D. Additional Results

Due to space limit, we place the full experiment results on all datasets/models in this section(see Tables 5 to 9 and Figures 9 and 10).

We further report additional semantic and entity-level evaluation metrics for both privacy leakage and model utility in Tables 12 and 13, including EntityRecovery-F1, BERTScore-F1 (Zhang et al., 2020), and BLEURT (Sellam et al., 2020), in addition to the BLEU/METEOR metrics adopted in the main paper for consistency with prior Split-LLM studies (Chen et al., 2024; Liu et al., 2025). EntityRecovery-F1 evaluates the recovery of sensitive entities from reconstructed text, while BERTScore and BLEURT measure semantic similarity. We additionally include a random baseline constructed from randomly sampled token sequences for reference. Overall, ADMI consistently achieves the lowest privacy leakage across lexical, semantic, and entity-level metrics, while maintaining competitive utility performance. These results demonstrate that the effectiveness of ADMI does not depend on a specific metric choice and generalizes beyond evaluation protocols.

*Table 13.* Additional utility metrics ($MP$).

| model | dataset | method | BertScore-F1 | BLEURT | METEOR(Origin) |
|---|---|---|---|---|---|
| Llama3.2-3B | Fin | w/o defense | 0.933 | 0.624 | 0.560 |
| Llama3.2-3B | Fin | ADMI | **0.924** | **0.589** | **0.516** |
| Llama3.2-3B | Fin | DualGuard | 0.915 | 0.547 | 0.459 |
| Llama3.2-3B | Fin | MID | 0.817 | 0.175 | 0.510 |
| Llama3.2-3B | Med | w/o defense | 0.867 | 0.352 | 0.176 |
| Llama3.2-3B | Med | ADMI | **0.863** | **0.340** | **0.173** |
| Llama3.2-3B | Med | DualGuard | 0.856 | 0.330 | 0.147 |
| Llama3.2-3B | Med | MID | 0.863 | 0.339 | 0.169 |
| Qwen2.5-7B | Fin | w/o defense | 0.930 | 0.614 | 0.550 |
| Qwen2.5-7B | Fin | ADMI | 0.924 | 0.591 | 0.527 |
| Qwen2.5-7B | Fin | DualGuard | **0.930** | **0.609** | **0.542** |
| Qwen2.5-7B | Fin | MID | 0.923 | 0.581 | 0.515 |
| Qwen2.5-7B | Med | w/o defense | 0.869 | 0.350 | 0.184 |
| Qwen2.5-7B | Med | ADMI | **0.867** | 0.345 | 0.182 |
| Qwen2.5-7B | Med | DualGuard | 0.867 | 0.345 | 0.178 |
| Qwen2.5-7B | Med | MID | 0.865 | **0.348** | **0.191** |

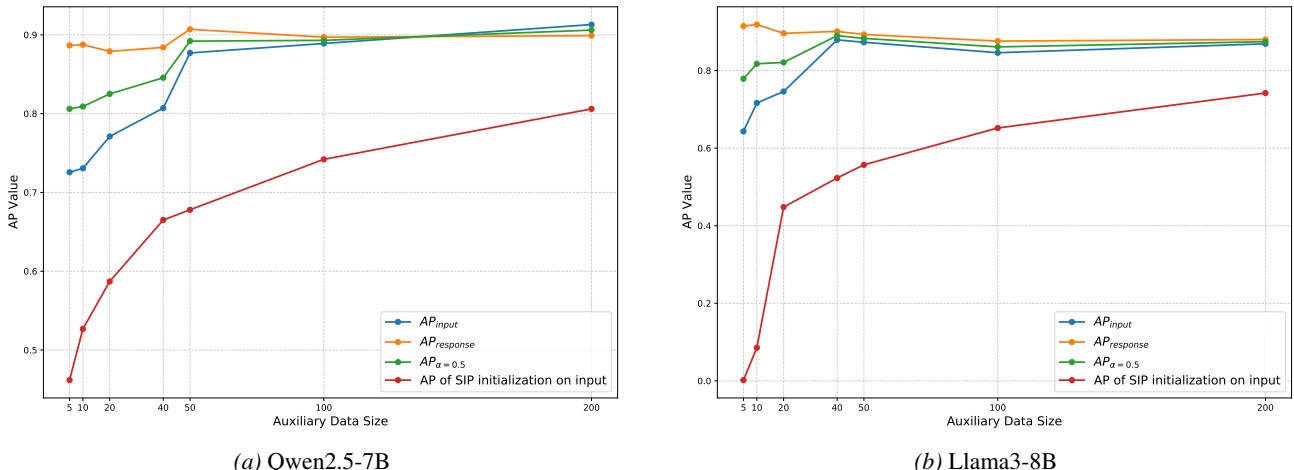

*(a)* Qwen2.5-7B                    *(b)* Llama3-8B

*Figure 11.* Impact of auxiliary dataset size on PIDI attack performance, evaluated on the Fin dataset.

