# OpenReview forum: "From Prompts to Responses: Dual-Sided Data Leakage and Defense in Split Large Language Models"
_ICML.cc/2026/Conference — ICML 2026 regular_

### Official Review · Reviewer_VaAL · 2026-03-07

**Soundness:** 3
**Presentation:** 3
**Significance:** 3
**Originality:** 3
**Overall Recommendation:** 5
**Confidence:** 3

**Summary:**

The paper identifies and addresses an emerging privacy vulnerability in split large language models (Split-LLM) under the "Head-Body-Tail (HBT)" configuration. While prior research has focused primarily on the leakage of private input prompts from the model head, this work explores a dual-sided leakage risk where the response outputs at the model tail are also susceptible to reconstruction by an honest-but-curious attacker. The authors propose Patched Model Inversion with Dual-Sided Initialization (PIDI), a two-stage attack involving dual-sided initialization and patched model inversion. A defense mechanism is also proposed termed Adapter-based DualGuard with Mutual Information Defense (ADMI). It leverages a mutual information regularizer to protect input prompts and a model distance regularizer to break the "not-too-far" property that enables response reconstruction.

**Compliance With Llm Reviewing Policy:**

Affirmed.

**Final Justification:**

The specific concerns I raised in my initial review have been addressed. However, I acknowledge that I am less familiar with the split-LLM literature than with the privacy literature, and so reviews from others with more expertise in split-LLM should be given greater weight when evaluating that aspect of the contributions.

**Key Questions For Authors:**

Please refer to the weaknesses.

**Limitations:**

Please include a discussion on potential imitations.

**Strengths And Weaknesses:**

**Strengths**
- **Problem formulation**: Systematically formalizes the dual-sided leakage (both prompts and responses) in Split-LLM generation, providing a more comprehensive threat model than existing head-only defenses.
- **Privacy-utility trade-off**: Empirical results across Llama and Qwen models show that ADMI significantly outperforms baselines including DualGuard and MID.

**Weaknesses**
- **Computational overhead**: The ADMI defense introduces additional complexity through the variational information bottleneck (VIB) and the adapter-based warmup. A more detailed analysis of the compute overhead would be useful.
- **Auxiliary data**: The attack's effectiveness may decrease if the attacker lacks domain-specific knowledge of the user's data.

---

> ### Author Rebuttal · Authors · 2026-03-30
>
> We sincerely thank the reviewer for the positive assessment of our problem formulation and constructive feedback.
>
> ## R1. Computational overhead
>
> | Model   | Method      | Inference Throughput (token/s/GPU) | Training Speed (s/iter) | Warmup Training Speed (s/iter) | Total Training Time (s) |
> | ---------- | ------------- | ---------------------------------- | ---------------------- | ----------------------------- | ---------------------- |
> | Qwen2.5-7B | w/o defense | 24.892  | 0.915 | /  | 13888.013 |
> | Qwen2.5-7B | ADMI | 20.418 | 1.381 | 0.449 | 21866.425 |
> | Llama3-8B | w/o defense | 23.354  | 0.996 | / | 17182.315 |
> | Llama3-8B | ADMI | 17.486 | 1.449 | 0.482 | 21843.745  |
>
>
> We evaluated the additional computational overhead introduced by our ADMI defense on the Dolly dataset. Our analysis yields the following observations:
>
> *   **Training Overhead:** The increase in total training time is primarily attributed to the inclusion of the local warm-up phase and the optimization of additional MID/Adapter models.
> *   **Inference Efficiency:** ADMI introduces a **moderate reduction in throughput** (approximately 18% for Qwen2.5-7B and 25% for Llama-8B).
>
> We acknowledge these trade-offs and will explicitly discuss them in the "Limitations" section of the revised manuscript.
>
> ## R2. Ablation on Auxiliary data size
> We acknowledge that in scenarios where the attacker lacks an auxiliary dataset (used for SIP initialization in PIDI), executing an attack on Split LLMs becomes significantly more challenging. To address this concern, we conducted additional ablation studies varying the size of the auxiliary data available to the attacker:
> https://anonymous.4open.science/r/ICML_rebuttal-3876/Qwen2.5-7B_aux_data_size.jpg
> https://anonymous.4open.science/r/ICML_rebuttal-3876/Llama3-8B_aux_data_size.jpg
>
> **Key Observations:**
> *   While reducing the auxiliary data visibly degrades the quality of SIP initialization, the final Attack Performance (AP) of PIDI remains stable once a minimal threshold is reached (~50 samples).
> *   This stability demonstrates the efficacy of PIDI's subsequent PMI stage, which effectively compensates for weaker initialization. This highlights the synergistic effect inherent in our design.
>
> These results indicate that PIDI is less sensitive to auxiliary data scarcity compared to SIP-only methods, requiring only a small data budget (50 samples) to maintain robust effectiveness.
>
> ## R3. Limitations
> We will explicitly incorporate a dedicated Limitations and Future Work section in the final manuscript to address the following:
>
> - While ADMI provides significant empirical privacy robustness, it introduces **additional architectural complexity** and results in a non-negligible training overhead and a moderate increase in inference latency.
> - Further, although the MI regularizer is grounded in an Information Bottleneck (IB) framework to minimize leakage bounds, it does not yet offer **certified privacy guarantee** for the end-to-end framework.
> - Future work should investigate in strategies for **improving efficiency** (e.g. pruning and quantization), and consider the possibility of even **stronger attackers** (e.g. who might be actively exploring the intermediate results and evolving during iterations).
>
>
> ## References
>
> - Chen et al. (2024). *Unveiling the Vulnerability of Private Fine-Tuning in Split-Based Frameworks for Large Language Models: A Bidirectionally Enhanced Attack*.

---

> > ### Author Rebuttal · Reviewer_VaAL · 2026-04-01
> >
> > Thank you for the response. The specific concerns I raised in my initial review have been addressed. However, I acknowledge that I am less familiar with the split-LLM literature than with the privacy literature, and so reviews from others with more expertise in split-LLM should be given greater weight when evaluating that aspect of the contributions.

---

> > > ### Author Response · Authors · 2026-04-06
> > >
> > > Thank you for your thoughtful follow-up and for taking the time to reassess our work. We sincerely appreciate your acknowledgment that the concerns raised in your initial review have been addressed. We are grateful for your careful evaluation and support.

---

### Official Review · Reviewer_7Q6U · 2026-03-09

**Soundness:** 3
**Presentation:** 3
**Significance:** 2
**Originality:** 3
**Overall Recommendation:** 5
**Confidence:** 1

**Summary:**

This paper addresses the double-ended leakage problem in Split-LLM (a real-world privacy-preserving fine-tuning/inference paradigm). The attack on PIDI improves reconstruction performance through double-sided initialization and piecewise inversion, while the defense against ADMI balances privacy and performance using adapter warm-up and mutual information regularization. These methods are logically sound and have no obvious vulnerabilities. Experiments on models such as Llama3/Qwen, across general/medical/financial datasets, using standard metrics such as BLEU/METEOR, show that ADMI's DCS score significantly exceeds the baseline.

**Compliance With Llm Reviewing Policy:**

Affirmed.

**Final Justification:**

Many thanks to the authors' effort!

My concerns about this paper remain somewhat unresolved. The main issue is my uncertainty about the extent of its contribution or benefit to the community as a whole. I've noticed that another reviewer, besides myself, expressed unfamiliarity with their research field. Furthermore, the previous work the authors followed appears to have only about 5 citations (I quickly read that paper, and at least in my opinion, it's not very practical), further suggesting that the domain studied in this paper may not currently be of much interest.

I maintain my rating and low confidence level. I hope the AC will consider the opinions of other reviewers who are more familiar with the domain.

**Key Questions For Authors:**

Please refer to the weaknesses part.

**Limitations:**

No.

Authors discuss the societal impact but do not discuss the limitations of their approaches, especially the defense part.
I suggest the authors include some.

**Strengths And Weaknesses:**

Strengths:

1. Novel and Practical Problem: This is the first systematic analysis of "two-sided leakage" (prompt + response) in the open generation of Split-LLM, filling the gap in head inversion and tail completion, which is particularly significant for highly sensitive scenarios such as finance and healthcare.

2. Innovative Attack Method: The two-sided initialization of PIDI (DSI, utilizing H and T activations) and patched inversion cleverly solve the problem of long sequence reconstruction. Experiments show a significant improvement in BLEU, outperforming baselines such as BiSR/VMI.

3. Practical and Effective Defense: ADMI, combined with local adapter warm-up (avoiding head freezing) and mutual information regularization, achieves DCS≈0.95+ on multiple models/datasets. It offers a better privacy-utility tradeoff than DualGuard/MID/DP, and is easily integrated into the open-source framework VFLAIR-LLM.

Weaknesses:

The analysis of the defense mechanism is superficial: although the effect is good, it does not theoretically prove why the adapter+MI combination is optimal (e.g., ablation only provides partial coverage), and the comparison with perturbation-based methods does not elaborate on the influence of the intensity parameter.

Minor:

I am more familiar with LLM privacy, but less so with split LLM. The authors could consider discussing LLM privacy-related work in the related work section to enhance the motivation for split LLM:
[1] Effective PII Extraction from LLMs through Augmented Few-Shot Learning.
[2] Reconstruct Your Previous Conversations! Comprehensively Investigating Privacy Leakage Risks in Conversations with GPT Models.
[3] Generated data with fake privacy: Hidden dangers of fine-tuning large language models on generated data.
[4] SOFT: Selective Data Obfuscation for Protecting LLM Fine-tuning against Membership Inference Attacks.

I'm unsure of the current and future impact of split LLM, so I'm giving it a 5 for now. If the split LLM domain doesn't achieve the expected impact, I might lower it to 4.

---

> ### Author Rebuttal · Authors · 2026-03-30
>
> We sincerely thank the reviewer for the constructive feedback.
>
> ## R1. Defense mechanism analysis
> Thank you for highlighting this point. While we acknowledge that "optimality" in complex, non-convex LLM landscapes is difficult to prove formally. Especially, providing a formal, end-to-end theoretical proof for the model tail's privacy upper bound is challenging due to the architectural complexity introduced by the Adapter Module and the heuristic $L_D$.
>
> Nevertheless, the MI regularizer is mathematically motivated by the Information Bottleneck (IB) principle. As derived below, **the MI term establishes a theoretical upper bound on information leakage through intermediate activations**:
> - Since $T$ is deterministic given $H'$, we have $I(X;H',T)=I(X;H')+I(X;T|H')=I(X;H')$.
> - Let $\hat X = \mathcal A(H',T)$ be any attacker-reconstructed input (deterministic or randomized). By data processing inequality, $I(X;\hat X) \le I(X;H',T) =I(X;H') \le L_{MI}$.
> - Thus, for any attacker reconstructed result $\hat X=\mathcal A(H',T)$,  $I(X;\hat X)\le I(X;H',T)\le L_{MI}$
>
>
> To move beyond "partial coverage", we have conducted additional ablation studies (semantic leakage/utility metrics, split-point sensitivity, auxiliary-data sensitivity) in our response to Reviewer Eahq (Q1/W3, W4).
> Our results consistently show that our framework achieves the highest privacy-utility trade-off in dual-sided leakage scenarios compared to other existing approaches. We will incorporate this deepened analysis and the expanded ablation results into the final manuscript.
>
>
> ## R2. Intensity Parameter Analysis
> We have already included an analysis of intensity parameters for the evaluated defenses in Figures 5 and 8. In these plots:
> - Each symbol represents the MP-AP performance of a specific method.
> - The **symbol size corresponds to the defense intensity**, controlled by the parameters listed in Table 3.
>
> To enhance clarity and facilitate interpretation, we will **increase the contrast in symbol sizes** and provide more descriptive captions in the revised manuscript.
>
> ## R3. Split-LLM impact and relation to broader LLM privacy work
> We appreciate the reviewer’s suggestion and will expand our Related Work section to better contextualize our contributions within the broader LLM privacy landscape. While many existing studies focus on centralized cloud deployment, our work is motivated by a critical **deployment-driven gap** in current privacy-preserving LLMs:
>
> Standard LLM deployment often forces a compromise between data privacy (compliance risks of external APIs), computational feasibility (the prohibitive cost of full local deployment), and model performance. While techniques such as Off-site Tuning [Xiao et al., 2023] and Knowledge Distillation [Hsieh et al., 2023] attempt to bridge this gap via compact local models, they frequently suffer from significant utility degradation.
>
> Consequently, **Split-LLM has emerged as a high-utility, resource-efficient Strategic Paradigm that is gaining significant traction** [Gu et al., 2025; Shen et al., 2023; Lin et al., 2024]. Within the Split-LLM ecosystem, prior research [Chen et al., 2024; Liu et al., 2025; Fu et al., 2022] has largely focused on isolated, single-sided leakage (either input or output). Our work addresses the more complex and realistic threat of dual-sided leakage in open-ended generation. By formalizing and defending this bidirectional channel, our work provides a critical, holistic security layer that is currently missing from the Split-LLM literature.
>
> ## R4. Limitations discussion
> We agree and will add an explicit discussion on limitations. Please refer to our response to Reviewer VaAL for details.
>
> ## References
> - Chen et al. (2024). *Unveiling the Vulnerability of Private Fine-Tuning in Split-Based Frameworks for Large Language Models: A Bidirectionally Enhanced Attack*.
> - Fu et al. (2022). *Label Inference Attacks Against Vertical Federated Learning*.
> - Gu et al. (2025). *VFLAIR-LLM: A Comprehensive Framework and Benchmark for Split Learning of LLMs*.
> - Lin et al. (2024). *SplitLoRA: A Split Parameter-Efficient Fine-Tuning Framework for Large Language Models*.
> - Liu et al. (2025). *DualGuard: A Parameter Space Transformation Approach for Bidirectional Defense in Split-Based LLM Fine-Tuning*.
> - Shen et al. (2023). *A Split-and-Privatize Framework for Large Language Model Fine-Tuning*
> - Xiao et al.(2023). *Offsite-Tuning: Transfer Learning without Full Model*
> - Hsieh et al.(2023). *Distilling Step-by-Step! Outperforming Larger Language Models with Less Training Data and Smaller Model Sizes*

---

> > ### Author Rebuttal · Reviewer_7Q6U · 2026-04-02
> >
> > I thank the authors for their rebuttal.
> >
> > Unfortunately, my questions remain unresolved. The work discussed in this paper does not seem to be of interest to the community at present (according to the reference and other reviewers' feedback). Furthermore, the methodology in this paper seems a bit similar to that in [1].
> >
> > However, I admit that I am unfamiliar with this topic. I am more familiar with work in the field of privacy. I hope that the AC will pay more attention to reviewers who are more familiar with this topic in the next steps.
> >
> > [1] Prompt Inversion Attack against Collaborative Inference of Large Language Models.

---

> > > ### Author Response · Authors · 2026-04-06
> > >
> > > ## R1. Clarification on the Importance of Split-LLM Privacy
> > >
> > > Thank you for the follow-up and for this important clarification. We understand your concern that Split-LLM privacy may currently be perceived as a relatively focused sub-area within the broader LLM privacy literature.
> > >
> > > At the same time, after carefully re-examining the cross-review comments, our understanding is that other reviewers did not express concerns regarding the importance of this research direction. On the contrary, their feedback appears to acknowledge its relevance within the broader LLM privacy landscape.
> > >
> > > Specifically, Reviewer Eahq notes that our problem formulation is relevant and describes extending Split-LLM privacy analysis to dual-sided leakage in open-ended generation as a **"natural and important step"**, and the problem framing **"is particularly relevant in domains such as finance and medicine"**.
> > > Reviewer VaAL and Reviewer x1j5 also highlight our problem formulation as strengths.
> > >
> > > Based on these observations, we respectfully interpret the cross-review feedback as suggesting that this research direction is meaningful. We further believe that this setting **is becoming increasingly relevant with the growing adoption of LLM-based services in privacy-sensitive areas**, making it important to understand the associated privacy risks in real-world deployments. Our goal is to highlight this emerging threat surface and provide a unified study, which we hope will help stimulate further research.
> > >
> > > ## R2. Clarifying the methodology is materially different
> > >
> > > Thank you for the follow-up and for pointing us to [1] (*Prompt Inversion Attack against Collaborative Inference of Large Language Models*). While both works consider optimization-based inversion attacks in split LLM settings, **the core problem scope and method targets are materially different**.
> > >
> > > As summarized in the table below, [1] is a one-sided prompt inversion study, while our work addresses a broader and different problem: unified dual-sided leakage (prompt + response) with a matched dual-sided defense pipeline. We will include [1] into the related work section and discuss the distinction.
> > >
> > > | Aspect                          | [1] Prompt Inversion Attack                                  | Ours                                                         |
> > > | ------------------------------- | ------------------------------------------------------------ | ------------------------------------------------------------ |
> > > | Privacy target                  | **Input prompt only**                                        | **Dual-sided leakage**: both private input $X$ and generated response $Y$ |
> > > | Leakage channel                 | Primarily one-side activation $A = F(E(x))$($H$ in our paper) from preceding layers | Both transmitted channels: head-side $H$ and tail-side $T$   |
> > > | Threat formulation              | attacker reconstructs **input prompt** during Split-LLM collaborative inference | attacker reconstructs **both prompt and response** during Split-LLM collaborative inference |
> > > | Meaning of "joint optimization" | Joint objective for **one-side prompt embedding** (activation matching + embedding-proximity constraint) | Joint refinement of the **full sequence (input + response)** after dual-sided initialization |
> > > | Initialization strategy         | Constrained optimization over prompt embedding (plus discretization calibration) | Dual-sided initialization: SIP on input + model completion response, then merged initialization $E_0$ |
> > > | Long-sequence handling          | no additional design                                         | Patched Model Inversion (patch-wise prefix refinement + final full inversion) |
> > > | Defense contribution            | Shows attacks can bypass existing defenses; no new defense method | Proposes ADMI ($L_{MI}$ + $L_D$ + adapter warm-up) for dual-sided defense |
> > >
> > >
> > >
> > > ## References
> > >
> > > [1] Prompt Inversion Attack against Collaborative Inference of Large Language Models.

---

### Official Review · Reviewer_Eahq · 2026-03-13

**Soundness:** 2
**Presentation:** 2
**Significance:** 2
**Originality:** 2
**Overall Recommendation:** 4
**Confidence:** 3

**Summary:**

This paper studies privacy leakage in Split-LLMs under a dual-sided threat model that includes both the input prompt and the generated response. The authors argue that prior work has focused primarily on input-side leakage from intermediate representations or on label leakage in classification settings, while prompt and response leakage in open-ended generation has not been jointly addressed. They consider an honest-but-curious model party that has access to the split model, the fine-tuned body, intermediate activations $H$ and $T$, and a small amount of auxiliary data, and formalize a setting in which both prompt and response are reconstructed during inference.

On the attack side, the paper proposes PIDI. The method first performs dual-sided initialization: SIP is used to initialize the prompt side, while model completion with a pretrained tail is used to initialize the response side. It then applies patched model inversion (PMI), which refines the reconstruction patch by patch before performing a final full inversion step. The stated motivation is that patch-wise inversion is more stable than vanilla inversion for long generated sequences. On the defense side, the paper proposes ADMI, which combines a mutual-information-based regularizer $L_{MI}$ to reduce leakage from head representations and a model-distance regularizer $L_D$ to weaken tail-side model completion attacks. The training pipeline consists of a local warm-up stage with adapters, followed by full defense training.

Experiments are conducted on three backbones (Llama3.2-3B, Llama3-8B, and Qwen2.5-7B) and three datasets (Fin, Med, and Dolly). In Table 1, PIDI attains the highest aggregate attack performance (AP) in most settings. Figures 4 and 6 present results across sequence lengths and show that patched inversion maintains stronger reconstruction performance than vanilla inversion as length increases. In Table 2, ADMI generally yields the lowest AP and the highest DCS among the compared defenses. The appendix also reports results for decomposed input/response leakage and model-completion-based attacks.

**Compliance With Llm Reviewing Policy:**

Affirmed.

**Final Justification:**

The paper identifies a meaningful problem — dual-sided privacy leakage in Split-LLM generation — and proposes both an attack (PIDI) and a defense (ADMI) supported by extensive experiments across three models and three datasets. The rebuttal addressed my main concerns: the random baseline comparison clarified that ADMI's residual BERTScore reflects the metric's inherent floor rather than meaningful semantic leakage, and the length-stratified analysis appropriately scoped PMI's contribution to larger models and longer sequences. With the framing in Section 7.1 revised accordingly and the length-stratified results incorporated as promised, the claims are well aligned with the evidence. I am raising my score to weak accept, as the contributions are solid and the experimental methodology is rigorous, though the practical strength of PMI is somewhat narrower than the original presentation suggested.

**Key Questions For Authors:**

Q1. Why are BLEU and METEOR sufficient metrics for privacy leakage and task utility in this setting? Have the authors considered semantic leakage metrics, sensitive entity recovery, or slot-level factual reconstruction metrics, and do the conclusions remain the same under those alternatives?

Q2. If PIDI is intended to be a strong dual-sided attack, how do the authors explain the appendix results in which DSI+VMI or MC+VMI are competitive with or better than PIDI on one side of the leakage problem? Is the main claim really about side-wise superiority, or only about the aggregate AP metric?

Q3. The main text emphasizes a favorable privacy-utility trade-off for ADMI, but Table 9 shows several settings with severe utility collapse under MC-based evaluation. How should these failure cases be interpreted?

**Limitations:**

yes

**Strengths And Weaknesses:**

S1. The problem formulation is relevant. Prior Split-LLM privacy work has indeed focused largely on input-side leakage or classification-style settings, so extending the discussion to dual-sided leakage in open-ended generation is a natural and important step. This framing is particularly relevant in domains such as finance and medicine, where the generated response itself may expose sensitive information.

S2. The paper identifies a practically meaningful failure mode of inversion on long sequences. The patch-wise refinement in PMI is not radically new, but it is a reasonable engineering response to the difficulty of reconstructing long generations with vanilla inversion. The length-based analysis in Figures 4 and 6 is aligned with this point.

---
W1. The methodological novelty appears more incremental and compositional than fundamental. PIDI combines SIP-based input initialization, model-completion-based response initialization, and a patched inversion refinement procedure, while ADMI combines a DualGuard-style training structure with an adapter module, a VIB-style mutual-information surrogate, and a model-distance regularizer. These integrations are sensible and empirically useful, but the paper’s novelty seems to lie primarily in system design and combination rather than in a genuinely new attack or defense principle.

W2. The claim of a "strong privacy guarantee" is not supported by the actual method. The abstract uses unusually strong language, but the paper does not provide a formal guarantee, certified bound, or theoretical privacy result. The $L_{MI}$ term is a variational surrogate, and $L_D$ is a heuristic regularizer that pushes the defended tail away from a pretrained one. This may support an empirical robustness claim, but it is not a privacy guarantee in the usual sense.

W3. The evaluation protocol is rather coarse for this problem. Privacy leakage is measured with BLEU, and utility is measured with METEOR. For open-ended generation, however, the relevant notion of leakage may be semantic recovery, sensitive-entity recovery, or factual slot reconstruction rather than n-gram overlap alone. Likewise, task utility is unlikely to be adequately captured by METEOR alone. Since DCS is also a composite metric built from these quantities, the overall conclusions depend substantially on the metric design.

W4. The experimental scope is somewhat limited for drawing strong robustness conclusions. The split point is fixed at $n_h = n_t = 4$, attack evaluation uses 200 test samples, and SIP uses 50 auxiliary samples. This is sufficient for a proof-of-concept, but it is not enough to establish robustness across deployment settings. The paper does not provide sensitivity analyses over split boundaries, attacker knowledge, or auxiliary data size.

W5. The defense story is significantly weakened by the appendix results under MC-based attacks. The main text presents ADMI as achieving strong privacy-utility trade-offs, but Table 9 shows multiple cases where utility collapses severely. For example, under MC defense evaluation, MP drops to 0 or near 0 in several Fin and Med settings, and even when utility is partially retained, response-side leakage can remain substantial. This is difficult to reconcile with the paper's repeated claim of "minimal impact on task performance." These failure cases should have been discussed much more directly in the main text.

---

> ### Author Rebuttal · Authors · 2026-03-30
>
> We sincerely thank the reviewer for the constructive feedback.
> ## W1. Novelty appears incremental
> We thank the reviewer for the opportunity to clarify our core contributions. Our novelty lies in both problem-setting and algorithmic design:
>
> **Problem Setting**: We are the first to formalize and investigate **dual-sided privacy leakage (input and response) in open-ended Split-LLM generation**, addressing critical gaps in head inversion and tail completion. This problem is particularly significant for highly sensitive scenarios such as finance and healthcare.
>
> **Algorithmic Innovations**:
> - PIDI jointly inverts inputs and responses by **leveraging dual leaked activation channels**. This synergy yields stable reconstruction quality unattainable by single-component attacks and remains robust even with minimal auxiliary data(see response to Reviewer VaAL).
> - PIDI's **patched inversion strategy** overcomes the "long-sequence" bottleneck where vanilla inversion fails.
> - ADMI **redesigns DualGuard's warm-up via an adapter path**, keeping the $M_b$ frozen yet active. This solution enhances training robustness while addressing DualGuard's architectural constraints, evidenced by its superior performance.
>
> ## W2. "Strong privacy guarantee" is over-claimed
> We appreciate the reviewer’s request for theoretical clarification. We acknowledge that providing a formal, end-to-end theoretical proof for the model's tail is challenging due to the architectural complexity introduced by the Adapter Module and the heuristic $L_D$. However, we can provide a formal proof that **the MI regularizer establishes a theoretical upper bound on information leakage through exposed channels**(following Zou et al., 2023; see response to Reviewer 7Q6U).
> we will revise our claims from "strong privacy guarantee" to "strong empirical privacy robustness" to avoid overclaiming.
>
> ## Q1 + W3. Are BLEU/METEOR sufficient?
> We appreciate the suggestion to broaden our evaluation. While our initial submission followed standard Split-LLM protocols (n-gram metrics) for comparability with [Chen et al., 2024; Liu et al., 2025; Gu et al., 2025], we have now added semantic (BERTScore/BLEURT) and entity-level (EntityRecovery-F1) metrics as presented here: https://anonymous.4open.science/r/ICML_rebuttal-3876/additional_metrics.jpg
>
> **The new results demonstrate that relative trends remain consistent with the original conclusion.** We will include this expanded analysis in the final revisions.
>
> ## Q2. Side-wise comparisons
> While DSI+VMI and MC+VMI match or exceed PIDI on $AP_{response}$ for Fin/Med datasets. This occurs because these datasets feature short sequences(see https://anonymous.4open.science/r/ICML_rebuttal-3876/Llama-3.2-3B_multi_dataset_comparison.jpg), where PIDI's advantage in long sequence is less pronounced.
>
> ## W4. Ablations: sample size, split point, auxiliary size
> (1) Attack sample size
>
> We conduct sensitivity analyses across varying attack evaluation sizes(see https://anonymous.4open.science/r/ICML_rebuttal-3876/llama3_dolly_attack_sample_size.PNG) and observed that performance stabilizes at `N=200`, justifying our choice for the main evaluation.
>
> (2) Split-point ablation
>
> We evaluated split points `3-3`, `5-5`, and `6-6` (besides `4-4`) on Llama3.2-3B over Fin Dataset under PIDI attack. Results are available in: https://anonymous.4open.science/r/ICML_rebuttal-3876/split_ablation.jpg
>
> With increasing $n_{h/t}$, $AP$ in w/o defense tends to degrade as a larger $M_{h/t}$ is harder to invert. While the protection performance of ADMI remains steady. We'll update results on other models if time permits.
>
> (3) Auxiliary data size
>
> Due to space limitations, Please refer to our response to Reviewer VaAL.
> ## Q3 + W5. utility-collapse concern in Table 8/9
> Thank you for catching this. We found a clerical error in the original column labels of Tables 8 and 9: the headers were incorrectly written as [$MP$, $AP_{response},AP_{input}$]], but should be [$AP_{input}$, $AP_{\alpha=0.5}$,$AP_{response}$](see corrected tables: https://anonymous.4open.science/r/ICML_rebuttal-3876/table8and9_with_correct_column_name.jpg).
>
> All numerical values and analyses remain unchanged—only the column names have been corrected. We sincerely apologize for this oversight and appreciate the reviewer’s diligence. We will ensure the final manuscript includes the corrected tables and consistent notation.
>
> ## References
> - Chen et al. (2024). *Unveiling the Vulnerability of Private Fine-Tuning in Split-Based Frameworks for Large Language Models: A Bidirectionally Enhanced Attack*.
> - Fu et al. (2022). *Label Inference Attacks Against Vertical Federated Learning*.
> - Gu et al. (2025). *VFLAIR-LLM: A Comprehensive Framework and Benchmark for Split Learning of LLMs*.
> - Liu et al. (2025). *DualGuard: A Parameter Space Transformation Approach for Bidirectional Defense in Split-Based LLM Fine-Tuning*.
> - Zou et al. (2023). *Mutual Information Regularization for Vertical Federated Learning*.

---

> > ### Author Rebuttal · Reviewer_Eahq · 2026-04-03
> >
> > I appreciate the authors' thorough rebuttal, particularly the corrected column labels for Tables 8/9 and the additional metric experiments. These efforts have fully the concern (W5) and substantially addressed W3. However, two follow-up points remain:
> >
> > (1) **PIDI's scope of advantage (Q2)**. The scatter plot demonstrates that PIDI's **advantage over DSI+VMI and BiSR diminishes considerably on shorter-sequence datasets** (Med dataset), where the three methods largely overlap. This suggests that PIDI's core contribution is **primarily the patched inversion mechanism for long sequences**, rather than a broadly superior dual-sided attack.  Yet the main text presents uniform dominance: Table 1 reports aggregate AP without length-stratified context, Figure 4 evaluates length robustness *only on Dolly* (the most favorable setting), and Section 7.1 claims *"PMI maintains stable and high AP across varying lengths"* without dataset qualification. How do the authors plan to revise this framing in the final manuscript?
> >
> > (2) **Absolute semantic leakage under ADMI (Q1)**. While the defense ranking trends are indeed consistent across metrics — ADMI remains the strongest defense — the **absolute BERTScore values after ADMI defense are notably high.** For example, on Llama3.2-3B / Fin, ADMI reduces BLEU from 0.868 to 0.017, yet BERTScore only drops from 0.977 to 0.753. This suggests that **non-trivial semantic similarity between reconstructed and original text persists even under defense.** Could the authors comment on whether this level of residual semantic leakage poses a practical privacy risk? A baseline reference (e.g., BERTScore between random unrelated sentence pairs) would help contextualize whether 0.753 represents meaningful leakage or simply reflects the high baseline similarity inherent to BERTScore.

---

> > > ### Author Response · Authors · 2026-04-06
> > >
> > > ## R1. PIDI's scope of advantage (Q2)
> > > We appreciate the reviewer’s detailed observation. We clarify that the observed overlap on the Med dataset does not indicate a fundamental weakness of PIDI, but rather reflects the specific operational regime of its components.  PIDI consists of two key modules: DSI (Dual-Sided Initialization) and PMI (Patched Model Inversion). Here, **DSI+VMI is a direct "without-PMI" ablation of PIDI**. Both methods share the exact same dual-sided initialization ($H$ and $T$), and the only difference is the inversion strategy—PMI in PIDI versus VMI in the ablation.
> > >
> > > To enable a clearer comparison of AP across different sequence lengths(which is difficult to observe in the original scatter plots), we conduct a more detailed **length-stratified (binned) AP analysis** in:
> > > https://anonymous.4open.science/r/ICML_rebuttal-3876/Llama-3.2-3B_multi_dataset_binned.jpg
> > > https://anonymous.4open.science/r/ICML_rebuttal-3876/Llama3-8B_multi_dataset_binned.jpg
> > > https://anonymous.4open.science/r/ICML_rebuttal-3876/Qwen2.5-7B_multi_dataset_binned.jpg
> > >
> > > The results reveal the following key insights:
> > > - The scenario where VMI outperforms PMI is strictly confined to **smaller models** (e.g., Llama-3.2-3B) with **short sequences (<100 tokens)**. In contrast, for larger models (Llama-3-8B, Qwen2.5-7B), PIDI significantly and consistently outperforms DSI+VMI across all sequence lengths.
> > >   - This indicates that PMI’s structured optimization becomes increasingly important as the embedding space grows more complex.
> > >   - Moreover, PMI is specifically designed for long-sequence optimization: it decomposes the full sequence into patches, performs prefix-consistent patch-wise inversion under the causal constraint, and then refines globally. When sequences are short (e.g., comparable to or shorter than the patch length $l=50$), **the number of effective patch stages is limited**. In small models—where inversion is intrinsically easier—**the marginal benefit of patch-wise refinement is correspondingly reduced**, allowing VMI to remain competitive.
> > > - The superior performance of DSI+VMI over BiSR(SIP+VMI) further demonstrates **the effectiveness of the DSI design**. Specifically, DSI leverages dual-channel information ($H$ and $T$) to perform role-aware initialization, whereas SIP considers only single-sided initialization using $H$. This dual-sided, role-aware strategy provides a more globally consistent and better-conditioned starting point for inversion, leading to improved reconstruction performance.
> > >
> > >
> > > **Revisions to the Manuscript**: Based on these findings, we will revise the main text to provide a more accurate framing:
> > > - Update length analysis & Section 7.1: We will add length-stratified results to the final manuscript and qualify the claim to: "PMI provides significant gains for long sequences and larger models, while its marginal benefit diminishes for short sequences in smaller models."
> > > - Clarify Adaptive Strategy: We will explicitly note that for short sequences, attackers can optimize efficiency by either increasing the budget for patch-stage optimization steps or bypassing the patching mechanism entirely (i.e., using the DSI+VMI branch). This positions PIDI as a flexible framework that adapts to sequence complexity.
> > >
> > > ## R2. Absolute semantic leakage under ADMI (Q1).
> > >
> > > To better contextualize these values, we introduce an additional **random baseline** in the report of additional AP metrics(see https://anonymous.4open.science/r/ICML_rebuttal-3876/additional_ap_with_random_baseline.png). Specifically, we construct this baseline by randomly sampling token sequences from the tokenizer vocabulary as reconstructed texts, and then compute the same set of metrics against the ground-truth references.
> > > We observe that:
> > > -  **The absolute BERTScore values are inherently high**, even for semantically unrelated text pairs (i.e., the random baseline consistently achieves a BERTScore around ~0.74).
> > > -  After applying ADMI, the **resulting BERTScore values are very close to this random baseline**.
> > > This indicates that ADMI can substantially mitigate information leakage.

---

### Official Review · Reviewer_x1j5 · 2026-03-14

**Soundness:** 2
**Presentation:** 2
**Significance:** 2
**Originality:** 2
**Overall Recommendation:** 3
**Confidence:** 3

**Summary:**

The paper identifies a vulnerability in Split-LLM through a Patched Model Inversion attack with Dual-Sided Initialization (PIDI), which targets both input prompts and output responses in the Split-LLM framework. The proposed attack combines dual-sided initialization with a patched inversion strategy to reconstruct sensitive information. To mitigate these threats, the paper further proposes Adapter-based DualGuard with Mutual Information Defense (ADMI), a defense mechanism designed to protect both the input and output sides of the system.

**Compliance With Llm Reviewing Policy:**

Affirmed.

**Key Questions For Authors:**

See Weaknesses

**Limitations:**

Yes

**Strengths And Weaknesses:**

## Strengths

1. The paper identifies a novel vulnerability in Split-LLM through Patched Model Inversion, which aims to reconstruct both the input prompt and the model response.
2. The paper further proposes a defense mechanism to mitigate the proposed attack.
3. Experimental results demonstrate the effectiveness of both the proposed attack and the corresponding defense method.

## Weaknesses

1. The paper mentions "traditional model inversion methods", but does not provide sufficient explanation or background.
2. The ``not-too-far'' property is introduced but not clearly explained.
3. There are minor formatting issues in the manuscript (e.g., a missing space before ``('').
4. The attack scenario is unclear. For example, users typically do not have access to the model $M_b$, while the model provider may not have access to $M_h$ and $M_t$. The assumed attacker capabilities should be clarified.
5. If the attackers are not involved in model training and do not have access to the full model, it is unclear how they can obtain a small amount of auxiliary data drawn from a distribution similar to that of the Data Party’s private dataset.

---

> ### Author Rebuttal · Authors · 2026-03-30
>
> We sincerely thank the reviewer for the constructive feedback.
>
> ## R1. Clarify "traditional model inversion methods"
> By "traditional model inversion", we refer to established inversion attacks [Fredrikson et al., 2015; Song & Raghunathan, 2020] that exploit information leakage from transmitted hidden states ($H = M_h(X)$) to infer the private input $X$.
>
> These attacks normally starts from an initial guess $X'$ and iteratively optimize it by minimizing an inversion loss $\mathcal{L}_{inv} = \|M_h(X') - H\|$, iteratively refining $X'$ to approach $X$. We will include this explanation in Section 4.
>
> In contrast, we propose **PIDI**, a novel inversion method specifically designed for the Split LLM open-ended generation setting. Unlike traditional approaches that focus solely on one side, PIDI **simultaneously inverts both the private input $X$ and the model response $Y$** by leveraging information from both activations $H$ and $T$ .
>
>
> ## R2. Clarify the "not-too-far" property
>
> The **"not-too-far" property**[Liu et al. (2025)] means fine-tuned LLM slices ($M_{h/t}$) remain relatively close to their pretrained counterparts($\bar M_{h/t}$), which can be exploited by model-completion style attacks [Liu et al., 2025]. Formally, it implies that the parameter shift $|\theta_{M_{h/t}} - \theta_{\bar M_{h/t}}|$ is very small. We will add a clearer definition in final paper revisions.
>
> ## R3. Clarify the attack scenario and attacker's capability
> Thank you for raising this point. We will make the threat model statement in Section 3 more explicit.
>
> As depicted in Figure 1, the process involves two distinct phases:
> - **Collaborative Fine-Tuning:** The Model Party and Data Party first collaboratively conduct Split LLM fine-tuning.
> - **Adversarial Inference:** Subsequently, during the collaborative inference phase, the Model Party (acting as the adversary) attempts to attack the Data Party to reconstruct private prompts and responses.
>
> **Adversary Capabilities (Model Party):**
> - As the model provider, it has the pretrained slices $\bar M_{h/b/t}$. It sends the pretrained slices $\bar M_{h/t}$ to data party for split fine-tuning, and therefore it owns fine-tuned body slice $M_b$, but not $M_{h/t}$.
> - It observes transmitted activations($H$ and $T$) during inference and tries to infer private data ($X$ and $Y$).
>
> **Victim Profile (Data Party):**
> - It performs local fine-tuning using provided pretrained slices $\bar{M}_{h/t}$.
> - During inference, it discloses sensitive information in its query input $X$ to generate model response $Y$, which also contains private information.
>
> ## R4. Auxiliary data assumption
>
> We appreciate your constructive feedback on the practical challenges of obtaining an in-distribution auxiliary private dataset. We acknowledge that in scenarios when the attacking party lacks an auxiliary dataset, executing an attack on Split LLMs becomes more difficult.
>
> To address this concern, we conducted additional **ablation studies on the size of the auxiliary data** available to the attacker (detailed in our response to Reviewer VaAL). Our results indicate that while attack performance degrades as the auxiliary dataset shrinks, the threat of PIDI remains **non-negligible even with limited data (e.g., 50 samples)**. We will incorporate this quantitative analysis into the final draft.
>
> Furthermore, we emphasize that investigating a "strong attacker" scenario—where a small amount of auxiliary data is accessible—allows us to design and validate more robust defense mechanisms. This conservative assumption ensures our proposed defenses remain effective under rigorous threat models.
>
> Notably, our auxiliary data setup aligns with established Split-LLM literature [Chen et al., 2024; Fu et al., 2022; Gu et al., 2025], which assumes the attacker possesses a limited in-distribution set to initialize and train attack models (e.g., SIP models). In practice, acquiring such a small volume of labeled data is feasible for an adversary. For example, if the adversary is a company, it can simply buy the labels from its employees.
>
> ## R5. Minor formatting issues
> Thank you for noticing this. We will carefully fix all spacing/formatting issues in the final revision.
>
> ## References
> - Chen et al. (2024). *Unveiling the Vulnerability of Private Fine-Tuning in Split-Based Frameworks for Large Language Models: A Bidirectionally Enhanced Attack*.
> - Fredrikson et al. (2015). *Model Inversion Attacks that Exploit Confidence Information and Basic Countermeasures*.
> - Fu et al. (2022). *Label Inference Attacks Against Vertical Federated Learning*.
> - Gu et al. (2025). *VFLAIR-LLM: A Comprehensive Framework and Benchmark for Split Learning of LLMs*.
> - Liu et al. (2025). *DualGuard: A Parameter Space Transformation Approach for Bidirectional Defense in Split-Based LLM Fine-Tuning*.
> - Song and Raghunathan (2020). *Information Leakage in Embedding Models*.

---

> > ### Author Rebuttal · Reviewer_x1j5 · 2026-04-02
> >
> > Thank you for your reply. The rebuttal partially addressed my concerns. Regarding the auxiliary data assumption, I am curious about how the method performs when the auxiliary data contain noise. In practice, regardless of how attackers obtain the auxiliary data, they cannot be certain whether these samples are truly drawn from the training distribution. For example, suppose an attacker obtains around 100 auxiliary samples, but some fraction of them is actually out-of-distribution. How would the method perform if 10%, 20%, or even 50% of the auxiliary data were noisy or out-of-distribution? Moreover, even if the attacker obtains such data, is there any practical way for them to identify and filter out samples that are not from the training distribution?

---

> > > ### Author Response · Authors · 2026-04-06
> > >
> > > ## R1. Auxiliary Data Contamination Ablation
> > > We performed an additional auxiliary data contamination stress test with a fixed auxiliary budget of 100 samples and various contamination ratios. We evaluated two contamination types:
> > > - **Random Token Noise**: Replacement with random token sequences. [Results](https://anonymous.4open.science/r/ICML_rebuttal-3876/Noise_Qwen_Llama.png)
> > > - **Cross-domain Mix**: Replacement of Finance samples(Fin Dataset) with general-domain (Alpaca Dataset) samples. [Results](https://anonymous.4open.science/r/ICML_rebuttal-3876/Mix_Qwen_Llama.png)
> > >
> > >
> > > **Observations**:
> > > - As contamination increases, SIP input reconstruction quality drops, while the final PIDI attack performance remains stable. This is consistent with our prior auxiliary-size sensitivity results(see our response to Reviewer VaAL). This demonstrates that **PIDI is robust to both the size and quality of auxiliary data, experiencing only mild performance degradation under high contamination**.
> > >   - The key reason is that auxiliary data is used solely for SIP initialization in PIDI’s first DSI stage, whereas **the second-stage inversion refinement can recover AP even when SIP initialization is imperfect**.
> > > - Divergent Impact of Contamination Types:
> > >   - SIP initialization relies on learning **an unsupervised inverse mapping from head-side representations to text tokens ($H \to X$) via cross-entropy loss**. This process fundamentally requires the model to identify statistical correlations between latent representations and token sequences.
> > >   - Random noise samles act as background noise without semantic structure, therefore leads to a threshold effect: SIP AP remains stable until contamination > 0.8, then collapses since extreme contamination leaves insufficient signal for the unsupervised inverse mapping ($H \to X$). Cross-domain sampls introduces a persistent bias rather than random chaos. It shifts the optimization trajectory but retains linguistic coherence and structural validity. This provides enough signal to maintain a functional mapping baseline, preventing the catastrophic collapse seen in high-ratio random noise.
> > >
> > >
> > > ## R2. Can attackers practically identify/filter OOD auxiliary data?
> > > We thank the reviewer for raising this important practical concern. First, we note that PIDI is inherently robust to auxiliary data quality, as shown in our prior contamination stress tests. This makes OOD contamination a less critical concern in practice.
> > >
> > > That said, in realistic settings, attackers can still reduce OOD contamination using widely available methods:
> > > - **Lightweight heuristic filtering** can serve as a cheap and scalable first pass. In many domains, in-distribution data exhibit distinctive structural or semantic patterns. Attackers can therefore eliminate clearly mismatched samples using simple rule-based criteria, such as format/template constraints and domain-specific lexical cues.
> > > - Attackers can further refine the auxiliary set using **model-based OOD scoring**, e.g., embedding samples into a feature space and filtering outliers via kNN or Mahalanobis distance, or one-class methods [Lee et al., 2018; Ruff et al., 2018]. Neighborhood-based membership inference [Mattern et al., 2023; Vahab et al. (2025)] can also serve as a filtering signal.
> > > We emphasize that OOD detection itself is a broad and well-studied research area, and a comprehensive treatment is beyond the scope of this work. Our goal here is not to propose new OOD detection techniques, but to highlight that practical and widely available tools already exist, which enable attackers to mitigate OOD noise to a certain extent.
> > >
> > > ## References
> > > - Ruff et al. (2018). *Deep One-Class Classification*.
> > > - Lee et al. (2018). *A Simple Unified Framework for Detecting Out-of-Distribution Samples and Adversarial Attacks*.
> > > - Mattern et al. (2023). *Membership Inference Attacks against Language Models via Neighbourhood Comparison*.
> > > - Vahab et al. (2025). *Inferring Private Data from Split and Partial Models: A Split Inference and Membership Inference Analysis via Knowledge Distillation*.

---

### Decision · Program_Chairs · 2026-04-30

**Decision:**

Accept (regular)

**Comment:**

This paper studies privacy risks in split large language model inference and introduces a dual-sided leakage formulation that considers both prompt reconstruction and response-side information exposure. It proposes a patched model inversion attack together with a mitigation strategy based on mutual-information regularization and adapter-based protection, and evaluates the approach on multiple datasets and models including LLaMA and Qwen.

The reviewers generally found the problem setting timely and the dual-sided perspective useful for understanding privacy risks in collaborative inference pipelines. The proposed attack–defense framework is supported by empirical evidence and provides a more complete treatment of leakage channels than prior work focusing only on prompt-side reconstruction. At the same time, some questions were raised about the assumptions behind the threat model, particularly the role of auxiliary data and the level of access required by the attacker. Reviewers also noted that the scope of the evaluation remains limited to the tested configurations and that comparisons with closely related inversion-style approaches could be further clarified.

The author responses helped explain the intended deployment scenario and clarified the distinction between prior one-sided inversion settings and the formulation studied here. While some limitations regarding generality remain, the paper makes a meaningful contribution to understanding privacy risks in split-LLM systems. Based on the reviewers’ assessments and my own evaluation, I recommend a weak accept.